# Collaborative Learning in the Jungle (Decentralized, Byzantine, Heterogeneous, Asynchronous and Nonconvex Learning)

**El-Mahdi El-Mhamdi** [*]
École Polytechnique
Palaiseau, France
`el-mahdi.el-mhamdi@polytechnique.edu`

**Sadegh Farhadkhani** [*]
IC School, EPFL
Lausanne, Switzerland
`sadegh.farhadkhani@epfl.ch`

**Rachid Guerraoui** [*]
IC School, EPFL
Lausanne, Switzerland
`rachid.guerraoui@epfl.ch`

**Arsany Guirguis** [*]
IC School, EPFL
Lausanne, Switzerland
`arsany.guirguis@epfl.ch`

**Lê-Nguyên Hoang** [*]
IC School, EPFL
Lausanne, Switzerland
`le.hoang@epfl.ch`

**Sébastien Rouault** [*]
IC School, EPFL
Lausanne, Switzerland
`sebastien.rouault@epfl.ch`

## Abstract

We study *Byzantine collaborative learning*, where $n$ nodes seek to collectively learn from each others' local data. The data distribution may vary from one node to another. No node is trusted, and $f < n$ nodes can behave arbitrarily. We prove that collaborative learning is equivalent to a new form of agreement, which we call *averaging agreement*. In this problem, nodes start each with an initial vector and seek to approximately agree on a common vector, which is close to the average of honest nodes' initial vectors. We present two asynchronous solutions to averaging agreement, each we prove optimal according to some dimension. The first, based on the minimum-diameter averaging, requires $n \geq 6f + 1$, but achieves asymptotically the best-possible averaging constant up to a multiplicative constant. The second, based on reliable broadcast and coordinate-wise trimmed mean, achieves optimal Byzantine resilience, i.e., $n \geq 3f + 1$. Each of these algorithms induces an optimal Byzantine collaborative learning protocol. In particular, our equivalence yields new impossibility theorems on what any collaborative learning algorithm can achieve in adversarial and heterogeneous environments.

## 1 Introduction

The distributed nature of data, the prohibitive cost of data transfers and the privacy concerns all call for collaborative machine learning. The idea consists for each machine to keep its data locally and to "simply" exchange with other machines what it learned so far. If all machines correctly communicate and execute the algorithms assigned to them, collaborative learning is rather easy. It can be achieved through the standard workhorse optimization algorithm: stochastic gradient descent (SGD) [37], which can be effectively distributed through averaging [26].

---

[*]Authors are listed alphabetically.

35th Conference on Neural Information Processing Systems (NeurIPS 2021).

But in a practical distributed setting, hardware components may crash, software can be buggy, communications can be slowed down, data can be corrupted and machines can be hacked. Besides, large-scale machine learning systems are trained on user-generated data, which may be crafted maliciously. For example, recommendation algorithms have such a large influence on social medias that there are huge incentives from industries and governments to fabricate data that bias the learning algorithms and increase the visibility of some contents over others [6, 33]. In the parlance of distributed computing, "nodes" can be *Byzantine* [28], i.e., they can behave arbitrarily maliciously, to confuse the system. Given that machine learning (ML) is now used in many critical applications (e.g., driving, medication, content moderation), its ability to tolerate Byzantine behavior is of paramount importance.

In this paper, we precisely define and address, for the first time, the problem of *collaborative learning* in a *fully decentralized*, *Byzantine*, *heterogeneous* and *asynchronous* environment with *non-convex* loss functions. We consider $n$ nodes, which may be machines or different accounts on a social media. Each node has its own local data, drawn from data distributions that may *greatly vary* across nodes. The nodes seek to collectively learn from each other, without however exchanging their data. *None of the nodes is trusted*, and any $f < n$ nodes can be Byzantine.

**Contributions.** We first precisely formulate the collaborative learning problem. Then, we give our main contribution: an equivalence between collaborative learning and a new more abstract problem we call *averaging agreement*. More precisely, we provide two reductions: from collaborative learning to averaging agreement and from averaging agreement to collaborative learning. We prove that both reductions essentially preserve the correctness guarantees on the output. The former reduction is the most challenging one to design and to prove correct. First, to update nodes' models, we use averaging agreement to aggregate nodes' stochastic gradients. Then, to avoid model drift, we regularly "contract" the nodes' models using averaging agreement. To prove correctness, we bound the diameter of honest nodes' models, and we analyze the *effective gradient* [12]. We then carefully select a halting iteration, for which correctness can be guaranteed.

Our tight reduction allows to derive both impossibility results and optimal algorithms for collaborative learning by studying the "simpler" averaging agreement problem. We prove lower bounds on the correctness and Byzantine resilience that any averaging agreement algorithm can achieve, which implies the same lower bounds for collaborative learning. We then propose two optimal algorithms for averaging agreement. Our first algorithm is asymptotically optimal with respect to correctness, up to a multiplicative constant, when nearly all nodes are honest. Our second algorithm achieves optimal Byzantine resilience. Each of these algorithms induces an optimal collaborative learning protocol.

While our algorithms apply in a very general setting, they can easily be tweaked for more specific settings with additional assumptions, such as the presence of a trusted parameter server [30], the assumption of homogeneous (i.e. i.i.d.) local data or synchrony (Section 3.3 and Section 6).

We implemented and evaluated our algorithms in a distributed environment with 3 ResNet models [24]. More specifically, we present their throughput overhead when compared to a non–robust collaborative learning approach with both i.i.d. and non–i.i.d. data (i.e. we highlight the cost of heterogeneity). Essentially, we show that our first algorithm is more lightweight with a slowdown of at most 1.7X in the i.i.d. case and almost the triple in the non–i.i.d. case. Our second algorithm adds slightly more than an order of magnitude overhead: here the non-i.i.d. slowdown is twice the i.i.d. one.

**Related work: Byzantine learning.** Several techniques have recently been proposed for Byzantine distributed learning, where different workers collaborate through a central *parameter server* [30] to minimize the average of their loss functions [26]. In each round, the server sends its model parameters to the workers which use their local data to compute gradients. Krum and Multi-Krum [4] use a distance–based scheme to eliminate Byzantine inputs and average the remaining ones. Median-based aggregation alternatives were also considered [38]. Bulyan [13] uses a meta–algorithm against a strong adversary that can fool the aforementioned aggregation rules in high–dimensional spaces. Coding schemes were used in Draco [8] and Detox [35]. In [2], quorums of workers enable to reach an information theoretical learning optimum, assuming however a strong convex loss function. Kardam [9] uses filters to tolerate Byzantine workers in an asynchronous setting. All these approaches assume a central *trusted* (parameter server) machine.

The few decentralized approaches that removed this single point of failure, restricted however the problem to (a) homogeneous data distribution, (b) convex functions, and/or (c) a weak (non–

Byzantine) adversary. MOZI [22] combines a distance–based aggregation rule with a performance–based filtering technique, assuming that adversaries send models with high loss values, restricting thereby the arbitrary nature of a Byzantine agent that can craft poisoned models whose losses are small only with respect to the honest nodes' incomplete loss functions. The technique is also inapplicable to heterogeneous learning, where nodes can have a biased loss function compared to the average of all loss functions[2]. BRIDGE [39] and ByRDiE [40] consider *gradient* descent (GD) and *coordinate* descent (CD) optimizations, respectively. Both rely on trimmed–mean to achieve Byzantine resilience assuming a synchronous environment and strongly convex loss functions[3] with homogeneous data distribution (*i.i.d.*). In addition, none of their optimization methods is stochastic: at each step, each node is supposed to compute the gradient on its entire local data set. ByzSGD [12] starts from the classical model of several workers and one server, which is then replicated for Byzantine resilience. It is assumed that up to 1/3 of the server replicas and up to 1/3 of the workers can be Byzantine, which is stronger than what we assume in the present paper where nodes play both roles and tolerate any subset of 1/3 Byzantine nodes. More importantly, ByzSGD assumes that all communication patterns between honest servers eventually hold with probability 1; we make no such assumption here. Additionally, our present paper is more general, considering heterogeneous data distributions, as opposed to [12]. Furthermore, heterogeneity naturally calls for *personalized* collaborative learning [14, 23, 10, 15], where nodes aim to learn local models, but still leverage collaborations to improve their local models. Interestingly, our general scheme encompasses personalized collaborative learning.

Maybe more importantly, our reduction to averaging agreement yields new more precise bounds that improve upon all the results listed above. These are we believe of interest, even in more centralized, homogeneous, synchronous and convex settings. In particular, our reduction can easily be adapted to settings where parameter servers and workers with local data are different entities, as in [12].

**Related work: Agreement.** A major challenge in collaborative learning is to guarantee "some" agreement between nodes about the appropriate parameters to consider. Especially in non-convex settings, this is critical as, otherwise, the gradients computed by a node may be completely irrelevant for another node. The agreement could be achieved using the traditional *consensus* abstraction [28]. Yet, consensus is impossible in asynchronous environments [18] and when it is possible (with partial synchrony), its usage is expensive and would be prohibitive in the context of modern ML models, with a dimension $d$ in the order of billions. In fact, and as we show in this paper, consensus is unnecessary.

An alternative candidate abstraction is *approximate agreement*. This is a weak form of consensus introduced in [11] where honest nodes converge to values that are close to each other, while *remaining in the convex hull* of the values proposed by honest nodes. In the one-dimensional case, optimal convergence rate has been achieved in both synchronous [16] and asynchronous environments [17], while optimal asynchronous Byzantine tolerance was attained by [1]. The multi-dimensional version was addressed by [31], requiring however $n^d$ local computations in each round, and assuming $n > f(d+2)$. This is clearly impractical in the context of modern ML.

By leveraging some distributed computing techniques [36, 1], we prove that collaborative learning can be reduced to *averaging agreement*, which is even weaker than approximate agreement. This enables us to bring down the requirement on the number of honest nodes from $n > f(d+2)$ to $n > 3f$, and only require linear computation time in $d$.

**Structure.** The rest of the paper is organized as follows. In Section 2, we precisely define the problems we aim to solve. Section 3 states our main result, namely, the equivalence between collaborative learning and averaging agreement. Section 4 describes our two solutions to averaging agreement, and proves their optimality. Section 5 reports on our empirical evaluation and highlight important takeaways. Finally, Section 6 concludes. The full proofs are provided in the supplementary material, as well as the optimized algorithm for homogeneous local data.

---

[2]Besides, MOZI, focusing on convex optimization, assumes that eventually, models on honest nodes do not drift among each others, which may not hold for Byzantine nodes could influence the honest models to drift away from each other [3].

[3]Convexity greatly helps, as the average of good models will necessarily be a good model. This is no longer the case in non-convex optimization, which includes the widely used neural network framework.

## 2 Model and Problem Definitions

### 2.1 Distributed computing assumptions

We consider a standard distributed computing model with a set $[n] = \{1, \ldots, n\}$ of nodes, out of which $h$ are honest and $f = n - h$ are Byzantine. For presentation simplicity, we assume that the first $h$ nodes are honest. But crucially, no honest node knows which $h - 1$ other nodes are honest. The $f$ Byzantine nodes know each other, can collude, and subsequently know who the $h$ remaining honest nodes are. Essentially, we assume a single adversary that controls all the Byzantine nodes. These nodes can send arbitrary messages, and they can send different messages to different nodes. In the terminology of distributed computing, the adversary is omniscient but not omnipotent. Such an adversary has access to all learning and deployment information, including the learning objective, the employed algorithm, as well as the dataset. We consider a general asynchronous setting [5]: the adversary can delay messages to honest nodes: no bound on communication delays or relative speeds is assumed. We denote BYZ the algorithm adopted by the adversary. Yet, we assume that the adversary is not able to delay all messages indefinitely [7]. Besides, the adversary is not able to alter the messages from the honest nodes, which can authenticate the source of a message to prevent spoofing and Sybil attacks.

Also for presentation simplicity, we assume that processes communicate in a round-based manner [5]. In each round, every honest node broadcasts a message (labelled with the round number) and waits until it successfully gathers messages from at most $q \leq h$ other nodes (labelled with the correct round number), before performing some local computation and moving to the next round. Even though the network is asynchronous, each round is guaranteed to eventually terminate for all honest nodes, as the $h$ honest nodes' messages will all be eventually delivered. Evidently, however, some of them may be delivered after the node receives $q$ messages (including Byzantine nodes'). Such messages will fail to be taken into account. Our learning algorithm will then rely on main rounds (denoted $t$ in Section 3), each of which is decomposed into sub-rounds that run averaging agreements.

### 2.2 Machine learning assumptions

We assume each honest node $j \in [h]$ has a local data distribution $\mathcal{D}_j$. The node's local loss function is derived from the parameters $\theta \in \mathbb{R}^d$, the model and the local data distribution, typically through $\mathcal{L}^{(j)}(\theta) = \mathbb{E}_{x \sim \mathcal{D}_j}[\ell(\theta, x)]$, where $\ell(\theta, x)$ is the loss for data point $x$, which may or may not include some regularization of the parameter $\theta$. Our model is agnostic to whether the local data distribution is a uniform distribution over collected data (i.e., empirical risk), or whether it is a theoretical unknown distribution the node can sample from (i.e., statistical risk). We make the following assumptions about this loss function.

**Assumption 1.** *The loss functions are non-negative, i.e., $\mathcal{L}^{(j)} \geq 0$ for all honest nodes $j \in [h]$.*

**Assumption 2.** *The loss functions are $L$-smooth, i.e., there exists a constant $L$ such that*

$$\forall \theta, \theta' \in \mathbb{R}^d, \ \forall j \in [h], \ \left\| \nabla \mathcal{L}^{(j)}(\theta) - \nabla \mathcal{L}^{(j)}(\theta') \right\|_2 \leq L \left\| \theta - \theta' \right\|_2. \tag{1}$$

**Assumption 3.** *The variance of the noise in the gradient estimations is uniformly bounded, i.e.,*

$$\forall j \in [h], \ \forall \theta \in \mathbb{R}^d, \ \mathbb{E}_{x \sim \mathcal{D}_j} \left\| \nabla_\theta \ell(\theta, x) - \nabla \mathcal{L}^{(j)}(\theta) \right\|_2^2 \leq \sigma^2. \tag{2}$$

*Moreover, the data samplings done by two different nodes are independent.*

**Assumption 4.** *There is a computable bound $\mathcal{L}_{max}$ such that, at initial point $\theta_1 \in \mathbb{R}^d$, for any honest node $j \in [h]$, we have $\mathcal{L}^{(j)}(\theta_1) \leq \mathcal{L}_{max}$.*

While the first three assumptions are standard, the fourth assumption deserves further explanation. Notice first that $\theta_1$ is a given parameter of our algorithms, which we could, just for the sake of the argument, set to $0$. The assumption would thus be about the value of the local losses at $0$, which will typically depend on the nodes' local data distribution. But losses usually depend on the data only as an average of the loss per data point. Moreover, the loss at $0$ for any data point is usually bounded. In image classification tasks for example, each color intensity of each pixel of an input image has a bounded value. This usually suffices to upper-bound the loss at $0$ for any data point, which then yields Assumption 4.

In iteration $t$, we require each node to average the stochastic gradient estimates over a batch of $b_t$ i.i.d. samples. As a result, denoting $\theta_t^{(j)}$ and $g_t^{(j)} \triangleq \frac{1}{b_t} \sum_{i \in [b_t]} \nabla_\theta \ell(\theta_t^{(j)}, x_{t,i}^{(j)})$ node $j$'s parameters and computed stochastic gradient in iteration $t$, we have

$$\mathbb{E}_{x \sim \mathcal{D}_j} \left\| g_t^{(j)} - \nabla \mathcal{L}^{(j)}(\theta_t^{(j)}) \right\|_2^2 \leq \frac{\sigma^2}{b_t} \triangleq \sigma_t^2. \tag{3}$$

As $t$ grows, we increase batch size $b_t$ up to $\Theta(1/\delta^2)$ (where $\delta$ is a parameter of the collaborative learning problem, see Section 2.3), so that $\sigma_t = \mathcal{O}(\delta)$ for $t$ large enough (see Remark 2). This allows to dynamically mitigate the decrease of the norm of the true gradient. Namely, early on, while we are far from convergence, this norm is usually large. It is then desirable to have very noisy estimates, as these can be obtained more efficiently, and as the aggregation of these poor estimates will nevertheless allow progress. However, as we get closer to convergence, the norm of the true gradient becomes smaller, making the learning more vulnerable to Byzantine attacks [3]. Increasing the batch size then becomes useful. Our proofs essentially formalize this intuition. Note that, in the homogeneous setting, [25] theoretically proves that leveraging Momentum reduces variance and prevents Byzantine attacks without increasing the batch size, while [32] observes on several hundred experimental settings that Momentum indeed often offsets, or even cancels out, the effects of Byzantine attacks.

## 2.3 Collaborative learning

Given the $\ell_2$ diameter $\Delta_2(\vec{\theta}) = \max_{j,k \in [h]} \left\| \theta^{(j)} - \theta^{(k)} \right\|_2$, collaborative learning consists in minimizing the average $\bar{\mathcal{L}}(\bar{\theta}) \triangleq \frac{1}{h} \sum_{j \in [h]} \mathcal{L}^{(j)}(\bar{\theta})$ of local losses at the average $\bar{\theta} \triangleq \frac{1}{h} \sum_{j \in [h]} \theta^{(j)}$, while guaranteeing that the honest nodes' parameters have a small diameter.

This general model encompasses to the *personalized* federated learning problem introduced by [14, 23, 10, 15]. For instance, in [23], each node $j$ aims to learn a local model $x_j$ that minimizes $f_j$, with a penalty $\frac{\lambda}{2} \left\| x_j - \bar{x} \right\|_2^2$ on their distance to the average $\bar{x}$ of all models. This framework can be restated by considering that nodes must agree on a common parameter $\theta = \bar{x}$, but have local losses defined by $\mathcal{L}^{(j)}(\theta) \triangleq \min_{x_j} f_j(x_j) + \frac{\lambda}{2} \left\| x_j - \theta \right\|_2^2$. The problem of [23] then boils down to minimizing the average of local losses.

**Definition 1.** *An algorithm* LEARN *solves the Byzantine $C$-collaborative learning problem if, given any local losses $\mathcal{L}^{(j)}$ for $j \in [h]$ satisfying assumptions (1,2,3,4) and any $\delta > 0$, no matter what Byzantine attack* BYZ *is adopted by Byzantines,* LEARN *outputs a vector family $\vec{\theta}$ of honest nodes such that*

$$\mathbb{E}\,\Delta_2(\vec{\theta})^2 \leq \delta^2 \quad and \quad \mathbb{E} \left\| \nabla \bar{\mathcal{L}}(\bar{\theta}) \right\|_2^2 \leq (1+\delta)^2 C^2 K^2, \tag{4}$$

*where $K \triangleq \sup_{j,k \in [h]}, \sup_{\theta \in \mathbb{R}^d} \left\| \nabla \mathcal{L}^{(j)}(\theta) - \nabla \mathcal{L}^{(k)}(\theta) \right\|_2$ is the largest difference between the true local gradients at the same parameter $\theta$, and where the randomness comes from the algorithm (typically the random sampling for gradient estimates).*

In our definition above, the constant $K$ measures the heterogeneity of local data distributions. Intuitively, this also captures the hardness of the problem. Indeed, the more heterogeneous the local data distributions, the more options Byzantine nodes have to bias the learning, the harder it is to learn in a Byzantine-resilient manner. Interestingly, for convex quadratic losses, our guarantee implies straightforwardly an upper-bound on the distance to the unique optimum of the problem, which is proportional to the hardness of the problem measured by $K$. In fact, our equivalence result conveys the tightness of this guarantee. In particular, the combination of our equivalence and of Theorem 5 implies that, for any $\varepsilon > 0$, asynchronous $(2f/h - \varepsilon)$-collaborative learning is impossible.

## 2.4 Averaging agreement

We address collaborative learning by reducing it to a new abstract distributed computing problem, which we call *averaging agreement*.

**Definition 2.** *A distributed algorithm* AVG *achieves Byzantine $C$-averaging agreement if, for any input $N \in \mathbb{N}$, any vector family $\vec{x} \in \mathbb{R}^{d \cdot h}$ and any Byzantine attack* BYZ*, denoting $\vec{y} \triangleq \mathrm{AVG}_N(\vec{x}, \mathrm{BYZ})$ the output of* AVG *given such inputs, we guarantee*

$$\mathbb{E}\,\Delta_2(\vec{y})^2 \leq \frac{\Delta_2(\vec{x})^2}{4^N} \quad and \quad \mathbb{E} \left\| \bar{y} - \bar{x} \right\|_2^2 \leq C^2 \Delta_2(\vec{x})^2, \tag{5}$$

where $\bar{y}_N = \frac{1}{h} \sum_{j \in [h]} y_N^{(j)}$ is the average of honest nodes' vectors, and where the randomness comes from the algorithm. We simply say that an algorithm solves averaging agreement if there exists a constant $C$ for which it solves $C$-averaging agreement.

In particular, for deterministic algorithms, $C$-averaging on input $N$ ensures the following guarantee

$$\Delta_2(\vec{y}) \leq \frac{\Delta_2(\vec{x})}{2^N} \quad \text{and} \quad \|\bar{y} - \bar{x}\|_2 \leq C\Delta_2(\vec{x}). \tag{6}$$

In Section 4, we will present two solutions to the averaging agreement problem. These solutions typically involve several rounds. At each round, each node sends their current vector to all other nodes. Then, once a node has received sufficiently many vectors, it will execute a robust mean estimator to these vectors, the output of which will be their starting vector for the next round. The nodes then halt after a number of rounds dependent on the parameter $N$.

# 3 The Equivalence

The main result of this paper is that, for $K > 0$, $C$-collaborative learning is equivalent to $C$-averaging agreement. We present two reductions, first from collaborative learning to averaging agreement, and then from averaging agreement to collaborative learning.

## 3.1 From collaborative learning to averaging agreement

Given an algorithm AVG that solves Byzantine $C$-averaging agreement, we design a Byzantine collaborative learning algorithm LEARN. Recall that LEARN must take a constant $\delta > 0$ as input, which determines the degree of agreement (i.e., learning quality) that LEARN must achieve.

All honest parameter vectors are initialized with the same random values (i.e., $\forall j \in [h], \theta_1^{(j)} = \theta_1$) using a pre-defined seed. At iteration $t$, each honest node $j \in [h]$ first computes a local gradient estimate $g_t^{(j)}$ given its local loss function $\mathcal{L}^{(j)}(\cdot)$ and its local parameters $\theta_t^{(j)}$, with a batch size $b_t$. But, instead of performing a learning step with this gradient estimate, LEARN uses an aggregate of all local gradients, which we compute using the averaging agreement algorithm AVG.

Recall from Definition 2 that AVG depends on a parameter which defines the degree of agreement. We set this parameter at $N(t) \triangleq \lceil \log_2 t \rceil$ at iteration $t$, so that $1/4^{N(t)} \leq 1/t^2$. Denoting $\vec{\gamma}_t$ the output of $\overrightarrow{\text{AVG}}_{N(t)}$ applied to vectors $\vec{g}_t$, we then have the following guarantee:

$$\mathbb{E}\,\Delta_2\left(\vec{\gamma}_t\right)^2 \leq \frac{\Delta_2\left(\vec{g}_t\right)^2}{t^2} \quad \text{and} \quad \mathbb{E}\,\|\bar{\gamma}_t - \bar{g}_t\|_2^2 \leq C^2 \Delta_2\left(\vec{g}_t\right)^2, \tag{7}$$

where the expectations are conditioned on $\vec{g}_t$. We then update node $j$'s parameters by $\theta_{t+1/2}^{(j)} = \theta_t^{(j)} - \eta \gamma_t^{(j)}$, for a fixed learning rate $\eta \triangleq \delta/12L$. But before moving on to the next iteration, we run once again AVG, with its parameter set to 1. Moreover, this time, AVG is run on local nodes' parameters. Denoting $\vec{\theta}_{t+1}$ the output of $\overrightarrow{\text{AVG}}_1$ executed with vectors $\vec{\theta}_{t+1/2}$, we then have

$$\mathbb{E}\,\Delta_2\left(\vec{\theta}_{t+1}\right)^2 \leq \frac{\Delta_2(\vec{\theta}_{t+1/2})^2}{4} \quad \text{and} \quad \mathbb{E}\left\|\bar{\theta}_{t+1} - \bar{\theta}_{t+1/2}\right\|_2^2 \leq C^2 \Delta_2(\vec{\theta}_{t+1/2})^2, \tag{8}$$

where the expectations are conditioned on $\vec{\theta}_{t+1/2}$. On input $\delta$, LEARN then runs $T \triangleq T_{\text{LEARN}}(\delta)$ learning iterations. The function $T_{\text{LEARN}}(\delta)$ will be given explicitly in the proof of Theorem 1, where we will stress the fact that it can be computed from the inputs of the problem[4] ($L$, $K$, $C$, $n$, $f$, $\sigma_t$, $\mathcal{L}_{max}$ and $\delta$). Finally, instead of returning $\vec{\theta}_{T_{\text{LEARN}}(\delta)}$, LEARN chooses uniformly randomly an iteration $* \in [T_{\text{LEARN}}(\delta)]$, using a predefined common seed, and returns the vector family $\vec{\theta}_*$. We recapitulate the local execution of LEARN (at each node) in Algorithm 1. We stress that on steps 6 and 8, when the averaging agreement algorithm AVG is called, the Byzantines can adopt any procedure BYZ, which

---

[4]Note that upper-bounds (even conservative) on such values suffice. Our guarantees still hold, though $T_{\text{LEARN}}(\delta)$ would take larger values, which makes LEARN slower to converge to $\vec{\theta}_*$.

consists in sending any message to any node at any point based on any information in the system, and in delaying for any amount of time any message sent by any honest node. Note that, apart from this, all other steps of Algorithm 1 is a purely local operation.

---

**Data:** Local loss gradient oracle, parameter $\delta > 0$
**Result:** Model parameters $\theta$

**1** Initialize local parameters $\theta_1$ using a fixed common seed;
**2** Fix learning rate $\eta \triangleq \delta/12L$;
**3** Fix number of iterations $T \triangleq T_{\text{LEARN}}(\delta)$;
**4** **for** $t \leftarrow 1, \dots, T$ **do**
**5** $\quad$ $g_t \leftarrow \text{GradientOracle}(\theta_t, b_t)$;
**6** $\quad$ $\gamma_t \leftarrow \text{AVG}_{N(t)}(\vec{g}_t, \text{BYZ})$ // Vulnerable to Byzantine attacks
**7** $\quad$ $\theta_{t+1/2} \leftarrow \theta_t - \eta\gamma_t$;
**8** $\quad$ $\theta_{t+1} \leftarrow \text{AVG}_1\left(\vec{\theta}_{t+1/2}, \text{BYZ}\right)$ // Vulnerable to Byzantine attacks
**9** **end**
**10** Draw $* \sim \mathcal{U}([T])$ using the fixed common seed;
**11** Return $\theta_*$;

**Algorithm 1:** LEARN execution on a honest node.

**Remark 1.** *In practice, it may be more efficient to return the last computed vector family, though our proof applies to a randomly selected iteration.*

**Theorem 1.** *Under assumptions (1, 2, 3, 4) and $K > 0$, given a $C$-averaging agreement oracle AVG, on any input $0 < \delta < 3$, LEARN solves Byzantine $C$-collaborative learning.*

The proof is quite technical, and is provided in the supplementary material. Essentially, it focuses on the average of all honest nodes' local parameters, and on the *effective gradients* that they undergo, given the local gradient updates and the applications of the averaging agreement oracle AVG.

**Remark 2.** *Our proof requires $T_{\text{LEARN}}(\delta) = \Theta\left(\delta^{-1}\max\left\{\delta^{-2}, (t \mapsto \sup_{s \geq t}\sigma_s)^{-1}(\delta)\right\}\right)$. To prevent the noise from being the bottleneck for the convergence rate, we then need $\sigma_t = \Theta(\delta)$, so that $T_{\text{LEARN}} = \Theta(\delta^{-3})$. Interestingly, this can be obtained by, for example, setting $b_t \triangleq t$, for $t \leq T_1 = \Theta(\delta^{-2})$, and $b_t \triangleq T_1$ for $t > T_1$, where $T_1$ is precisely defined by the proof provided in the supplementary material. In particular, we do not need to assume that $b_t \to \infty$.*

## 3.2 Converse reduction

We also prove the converse reduction, from averaging agreement to collaborative learning.

**Theorem 2.** *Given a Byzantine $C$-collaborative learning oracle, then, for any $\delta > 0$, there is a solution to Byzantine $(1 + \delta)C$-averaging agreement.*

The proof is given in the supplementary material. It is obtained by applying a collaborative learning algorithm LEARN to the local loss functions $\mathcal{L}^{(j)}(\theta) \triangleq \left\|\theta - x^{(j)}\right\|_2^2$.

This converse reduction proves the tightness of our former reduction, and allows to straightforwardly derive impossibility theorems about collaborative learning from impossibility theorems about averaging agreement. In particular, in the sequel, we prove that no asynchronous algorithm can achieve better than Byzantine $\frac{h+2f-q}{h}$-averaging agreement. It follows that no algorithm can achieve better than Byzantine $\frac{h+2f-q}{h}$-collaborative learning. Similarly, Theorem 6 implies the impossibility of Byzantine asynchronous collaborative learning for $n \leq 3f$.

## 3.3 Particular cases

**Trusted server.** It is straightforward to adapt our techniques and prove the equivalence between averaging agreement and collaborative learning in a context with a trusted server. Our lower bounds for asynchronous collaborative learning still apply to $C$-averaging agreement, and thus to $C$-collaborative learning. Note, however, that the trusted server may allow to improve the speed of collaborative learning, as it no longer requires contracting the parameters of local nodes' models.

**Homogeneous learning.** In the supplementary material, we propose a faster algorithm for i.i.d. data, called HOM-LEARN, which skips the averaging agreement of nodes' gradients. Despite requiring fewer communications, HOM-LEARN remains correct, as the following theorem shows.

**Theorem 3.** *Under assumptions (1, 2, 3, 4), for i.i.d. local data and given a C-averaging agreement oracle AVG, on input $\delta > 0$, HOM-LEARN guarantees $\mathbb{E}\,\Delta_2(\vec{\theta}_*)^2 \leq \delta^2$ and $\mathbb{E}\left\|\nabla\mathcal{L}\left(\bar{\theta}_*\right)\right\|_2^2 \leq \delta^2$.*

## 4 Solutions to Averaging Agreement

We now present two solutions to the averaging agreement problem, called MINIMUM–DIAMETER AVERAGING[5] (MDA) and RELIABLE BROADCAST - TRIMMED MEAN (RB-TM), each thus inducing a solution to collaborative learning. We prove each optimal according to some dimension.

### 4.1 Optimal averaging

Given a family $\vec{z} \in \mathbb{R}^{d \cdot q}$ of vectors, MDA first identifies a subfamily $S_{\mathrm{MDA}}(\vec{z})$ of $q - f$ vectors of minimal $\ell_2$ diameter, i.e.,

$$S_{\mathrm{MDA}}(\vec{z}) \in \underset{\substack{S \subset [q] \\ |S| = q-f}}{\arg\min}\, \Delta_2\left(\vec{z}^{(S)}\right) = \underset{\substack{S \subset [q] \\ |S| = q-f}}{\arg\min}\, \max_{j,k \in S} \left\| z^{(j)} - z^{(k)} \right\|_2. \qquad (9)$$

We denote $\vec{z}^{(\mathrm{MDA})} \in \mathbb{R}^{d \cdot (q-f)}$ the subfamily thereby selected. MDA then outputs the average of this subfamily, i.e.,

$$\mathrm{MDA}(\vec{z}) \triangleq \frac{1}{q-f} \sum_{j \in S_{\mathrm{MDA}}(\vec{z})} z^{(j)}. \qquad (10)$$

On input $N \in \mathbb{N}$, $\mathrm{MDA}_N$ iterates MDA $T_{\mathrm{MDA}}(N) = \lceil N \ln 2/\tilde{\varepsilon} \rceil$ times on vectors received from other nodes at each communication round such that the output of round $t$ will be the input of round $t + 1$. The correctness of MDA is then ensured under the following assumption.

**Assumption 5** (Assumption for analysis of MDA). *There is $0 < \varepsilon < 1$ such that $n \geq \frac{6+2\varepsilon}{1-\varepsilon} f$. This then allows to set $q \geq \frac{1+\varepsilon}{2} h + \frac{5+3\varepsilon}{2} f$. In this case, we define $\tilde{\varepsilon} \triangleq \frac{2\varepsilon}{1+\varepsilon}$.*

**Theorem 4.** *Under Assumption 5, MDA achieves Byzantine $\frac{(2f+h-q)q+(q-2f)f}{h(q-f)\tilde{\varepsilon}}$-averaging agreement.*

The proof is given in the supplementary material. It relies on the observation that, because of the filter, no Byzantine vector can significantly harm the estimation of the average.

**Remark 3.** *Although MDA runs in linear time in $d$, it runs in exponential time in $q$. Interestingly, assuming that each honest node fully trusts its computations and its data (which may not hold if parameter-servers do not compute gradients as in [30]), each honest node can use its own vector to filter out the $f$ most dissimilar gradients in linear time in $q$, and can average out all remaining vectors. Using a similar proof as for MDA, the algorithm thereby defined can be shown to achieve asymptotically the same averaging constant as MDA, in the limit $q \gg f$; but it now runs in $\mathcal{O}(dq)$, (requiring however $n \geq 7f + 1$).*

**Theorem 5.** *No asynchronous algorithm can achieve better than Byzantine $\frac{h+2f-q}{h}$-averaging agreement.*

The proof is given in the supplementary material. It relies on the quasi-unanimity lemma, which shows that if a node receives at least $q - f$ identical vectors $x$, then it must output $x$. We then construct an instance where, because of this and of Byzantines, honest nodes cannot agree. Note that, as a corollary, in the regime $q = h \to \infty$ and $f = o(h)$, MDA achieves asymptotically the best-possible averaging constant, up to a multiplicative constant equal to $3/2$.

---

[5]Introduced by [13] in the context of robust machine learning, it uses the same principle as the minimal volume ellipsoid, that was introduced by [36] in the context of robust statistics.

## 4.2 Optimal Byzantine resilience

Our second algorithm makes use of reliable broadcast[6]: each Byzantine node broadcasts only a single vector (the uniqueness property of reliable broadcast in [1]). We denote $\vec{w} \in \mathbb{R}^{d \cdot n}$ the family of vectors proposed by all nodes. For each $j \in [h]$, $w^{(j)} = x^{(j)}$ is the vector of an honest node, while a Byzantine node proposes $w^{(j)}$ for each $j \in [h+1, n]$. Moreover, [1] showed the existence of a multi-round algorithm which, by using reliable broadcast and a witness mechanism, guarantees that any two honest nodes $j$ and $k$ will collect at least $q$ similar inputs. Formally, denoting $Q^{(j)} \subseteq [n]$ the set of nodes whose messages were successfully delivered to node $j$ (including through relays), the algorithm by [1] guarantees that $\left| Q^{(j)} \cap Q^{(k)} \right| \geq q$ for any two honest nodes $j, k \in [h]$. At each iteration of our RB-TM algorithm, each node $j$ exploits the same reliable broadcast and witness mechanism techniques to collect other nodes' vectors. Now, given its set $Q^{(j)}$ of collected messages, each node $j$ applies coordinate-wise trimmed mean, denoted TM, as follows. For each coordinate $i$, it discards the $f$ smallest $i$-th coordinates it collected, as well as the $f$ largest. We denote $\vec{z}^{(j)} = \vec{w}^{(Q^{(j)})}$ the subfamily received by node $j$, and $S(\vec{z}^{(j)}[i]) \subset [n]$ the subset of nodes whose $i$-th coordinates remain after trimming. Node $j$ then computes the average $y^{(j)}$ of the $i$-th coordinates of this subset, i.e.

$$y^{(j)}[i] \triangleq \frac{1}{\left| Q^{(j)} \right| - 2f} \sum_{k \in S(\vec{z}^{(j)}[i])} w^{(k)}[i]. \tag{11}$$

RB-TM consists of iterating TM, on vectors received from other nodes at each communication round. Namely, given input $N \in \mathbb{N}$, RB-TM iterates TM $T_{\text{RB-TM}}(N)$ times, where

$$T_{\text{RB-TM}}(N) \triangleq \left\lceil \frac{(N+1)\ln 2 + \ln \sqrt{h}}{\tilde{\varepsilon}} \right\rceil. \tag{12}$$

The correctness of RB-TM can then be guaranteed under the following assumption.

**Assumption 6** (Assumption for analysis of RB-TM). *There is $\varepsilon > 0$ such that $n \geq (3 + \varepsilon)f$. We then set $q = n - f$, and define $\tilde{\varepsilon} \triangleq \frac{\varepsilon}{1+\varepsilon}$.*

**Theorem 6.** *Under Assumption 6, RB-TM guarantees Byzantine $\frac{4f}{\sqrt{h}}$-averaging agreement. This is optimal in terms of Byzantine resilience. Indeed, for $n \leq 3f$, no algorithm can achieve Byzantine averaging agreement.*

The proof is provided in the supplementary material. The correctness relies on a coordinate-wise analysis, and on the study of a so-called *coordinate-wise diameter*, and its relation with the $\ell_2$ diameter. The lower bound exploits the quasi-unanimity lemma. Note that while RB-TM tolerates more Byzantine nodes, its averaging constant is larger than that of MDA by a factor of $\mathcal{O}(\sqrt{h})$.

## 5 Empirical Evaluation

We implemented our collaborative learning algorithms using *Garfield* library [20] and PyTorch [34]. Each agreement algorithm comes in two variants: one assuming i.i.d. data (See supplementary material) and one tolerating non-i.i.d. data (Algorithm 1). In each case, the first variants require fewer communications. We report below on the empirical evaluation of the overhead of our four variants when compared to a non–robust collaborative learning approach. Our baseline is indeed a vanilla fully decentralized implementation in which all nodes share their updates with each other and then aggregate these updates by *averaging* (a deployment that cannot tolerate even one Byzantine node).

We focus on throughput, measuring the number of updates the system performs per second. As we consider an asynchronous network, we report on the fastest node in each experiment. We consider image classification tasks, using MNIST [29] and CIFAR-10 [27] datasets. MNIST is a dataset of handwritten digits with 70,000 $28 \times 28$ images in 10 classes. CIFAR-10 consists of 60,000 $32 \times 32$ colour images in 10 classes. We use batches of size 100, and we experimented with 5 models with different sizes ranging from simple models like small convolutional neural network (*MNIST_CNN* and *Cifarnet*), training a few thousands of parameters, to big models like ResNet-50 with around 23M

---

[6]Note that MDA can also be straightforwardly upgraded into RB-MDA to gain Byzantine resilience.

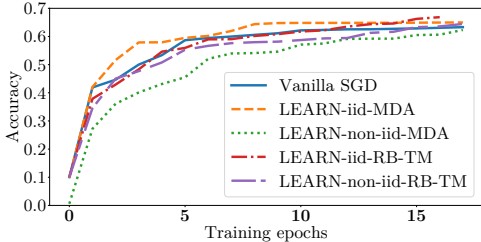

Figure 1: Convergence of our algorithms and the vanilla baseline.

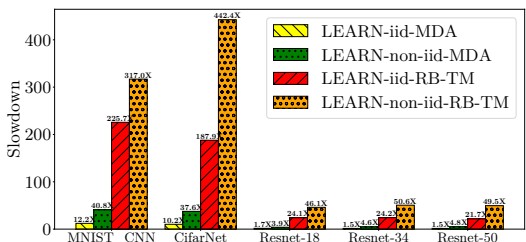

Figure 2: Slowdown of our algorithms normalized to the vanilla baseline throughput.

parameters. Our experimental platform is Grid5000 [19]. We always employ nodes from the same cluster, each having 2 CPUs (Intel Xeon E5-2630 v4) with 14 cores, 768 GiB RAM, 2×10 Gbps Ethernet, and 2 Nvidia Tesla P100 GPUs. We set $f = 1$, except when deploying our vanilla baseline.

Figure 1 compares the convergence of our algorithms to the vanilla baseline w.r.t. the training epochs. We use 7 nodes in this experiment, and we train Resnet–18 with CIFAR10. We verify from this figure that our algorithms can follow the same convergence trajectory as the vanilla baseline. It is clear from the figure that the *i.i.d.* versions outperform the *non-i.i.d.* ones.

Figure 2 depicts the throughput overhead of our algorithms (with both i.i.d. and non-i.i.d. data) compared to our vanilla baseline, with 10 nodes from the same cluster. Three observations from this figure are in order. First, the MDA–based algorithm performs better than the RB-TM one. The reason is that the latter incurs much more communication messages than the former as the latter uses reliable broadcast and a witness mechanism. Second, tolerating Byzantine nodes with i.i.d. data is much cheaper than the non-i.i.d. case. The reason is that it is harder to detect Byzantine behavior when data is not identically distributed on the nodes, which translates into more communication steps. Third, the slowdown is much higher with small models (i.e., *MNIST_CNN* and *Cifarnet*). This is because the network bandwidth is not saturated by the small models in the vanilla case, where it gets congested with the many communication rounds required by our algorithms. On the other hand, with the larger models, the vanilla deployment saturates the network bandwidth, making the extra communication messages account only for linear overhead.

Finally, it is important to notice that our evaluation is by no means exhaustive and our implementation has not been optimized. Our goal was to give an overview of the relative overheads. With proper optimizations, we believe the actual throughput could be increased for all implementations.

## 6   Conclusion

We defined and solved collaborative learning in a fully decentralized, Byzantine, heterogeneous, asynchronous and non-convex setting. We proved that the problem is equivalent to a new abstract form of agreement, which we call averaging agreement. We then described two solutions to averaging agreement, inducing two original solutions to collaborative learning. Each solution is optimal along some dimension. In particular, our lower bounds for the averaging agreement problem provide lower bounds on what any collaborative learning algorithm can achieve. Such impossibility results would have been challenging to obtain without our reduction. Our algorithms and our impossibility theorems are very general but can also be adapted for specific settings, such as the presence of a trusted parameter server, the assumption of i.i.d. data or a synchronous context[7]. In the latter case for instance, our two algorithms would only require $n \geq 4f + 1$ and $n \geq 2f + 1$, respectively.

**Limitations and potential negative social impacts.** Like all Byzantine learning algorithms, we essentially filter out outliers. In practice, this may discard minorities with vastly diverging views. Future research should aim to address this fundamental trade-off between inclusivity and robustness. We also note that the computation time of MDA grows exponentially with $q$, when $f$ is a constant fraction of $q$.

---

[7]In the synchronous case of MDA, with $q = n$, $n \geq \frac{4+2\varepsilon}{1-\varepsilon} f$ is sufficient to guarantee $q \geq \frac{1+\varepsilon}{2} h + \frac{5+3\varepsilon}{2} f$ in Assumption 5. Also, note that no synchronous algorithm can achieve better than $\frac{f}{h}$-averaging agreement.

## Acknowledgments and Disclosure of Funding

We thank Rafaël Pinot and Nirupam Gupta for their useful comments. This work has been supported in part by the Swiss National Science Foundation projects: 200021_182542, Machine learning and 200021_200477, Controlling the spread of Epidemics. Most experiments presented in this paper were carried out using the Grid'5000 testbed, supported by a scientific interest group hosted by Inria and including CNRS, RENATER and several Universities as well as other organizations (see `https://www.grid5000.fr`).

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
