# Supplementary Material

In the proofs, we call the first and second properties of averaging agreement "asymptotic agreement" and "C-averaging" guarantee respectively (see Equation (5) in the paper).

Moreover, we denote by $\overrightarrow{\text{BYZ}}^{(j)}(\vec{x}) = \left( \text{BYZ}^{(j,1)}(\vec{x}), \ldots, \text{BYZ}^{(j,q)}(\vec{x}) \right)$ the family of inputs (of size $q$) collected by node $j$. There thus exists a bijection $\tau : Q_t^{(j)} \to [q]$, where $Q_t^{(j)}$ is the set of nodes that successfully delivered messages to $j$ at round $t$, such that $\text{BYZ}^{(j,\tau(k))} = x^{(k)}$ for all honest nodes $k \in [h]$.

## A  The Equivalence

**Preliminary lemmas**

**Lemma 1.** *For any $\alpha > 0$ and any two vectors $u$ and $v$, we have*

$$\|u + v\|_2^2 \le (1 + \alpha^{-1}) \|u\|_2^2 + (1 + \alpha) \|v\|_2^2. \tag{13}$$

*As an immediate corollary, for any two families $\vec{u}$ and $\vec{v}$ of vectors, we have*

$$\Delta_2 (\vec{u} + \vec{v})^2 \le (1 + \alpha^{-1}) \Delta_2 (\vec{u})^2 + (1 + \alpha) \Delta_2 (\vec{v})^2. \tag{14}$$

*Proof.* We have the following inequalities:

$$(1 + \alpha^{-1}) \|u\|_2^2 + (1 + \alpha) \|v\|_2^2 - \|u + v\|_2^2 = \alpha^{-1} \|u\|_2^2 + \alpha \|v\|_2^2 - 2u \cdot v \tag{15}$$

$$= \left\| \alpha^{-1/2} u - \alpha^{1/2} v \right\|_2^2 \ge 0. \tag{16}$$

Rearranging the terms yields the lemma. $\qquad\square$

**Lemma 2.** *For any vector family $u_1, \ldots, u_N$, we have*

$$\left\| \sum_{j \in [N]} u_j \right\|_2^2 \le N \sum_{j \in [N]} \|u_j\|_2^2. \tag{17}$$

*As an immediate corollary, for any family of vector families $\vec{u_1}, \ldots, \vec{u_N}$, we have*

$$\Delta_2 \left( \sum_{j \in [N]} \vec{u_j} \right)^2 \le N \sum_{j \in [N]} \Delta_2 (\vec{u_j})^2. \tag{18}$$

*Proof.* Notice that $u \mapsto \|u\|_2^2$ is a convex function. As a result,

$$\left\| \frac{1}{N} \sum_{j \in [N]} u_j \right\|_2^2 \le \frac{1}{N} \sum_{j \in [N]} \|u_j\|_2^2. \tag{19}$$

Multiplying both sides by $N^2$ allows to conclude. $\qquad\square$

**Lemma 3.** *For any vector family $\vec{u} \in \mathbb{R}^{d \cdot h}$, we have*

$$\Delta_2(\vec{u}) \le 2 \max_{j \in [h]} \left\| u^{(j)} \right\|_2. \tag{20}$$

*Proof.* We have the inequalities

$$\Delta_2(\vec{u}) = \max_{j,k \in [h]} \left\| u^{(j)} - u^{(k)} \right\|_2 \le \max_{j,k \in [h]} \left\| u^{(j)} \right\|_2 + \left\| u^{(k)} \right\|_2 \tag{21}$$

$$= \max_{j \in [h]} \left\| u^{(j)} \right\|_2 + \max_{k \in [h]} \left\| u^{(k)} \right\|_2 = 2 \max_{j \in [h]} \left\| u^{(j)} \right\|_2, \tag{22}$$

which is the lemma. $\qquad\square$

We now prove that LEARN solves collaborative learning. Note that all the proofs depend on some quantity $\alpha_t$, which will eventually be defined as $\alpha_t \triangleq \max\left\{1/\sqrt{t}, \sigma_t\right\}$. Note also that we then have $\alpha_t \leq \bar{\alpha} \triangleq \max\{1, \sigma\}$ since $b_t \geq 1$. We also define

$$\xi_t^{(j)} \triangleq g_t^{(j)} - \nabla\mathcal{L}^{(j)}(\theta_t^{(j)}), \tag{23}$$

the gradient estimation error (noise) of node $j$ at round $t$, whose norm is bounded by $\sigma_t$ (see Section 2.2).

**Lemma 4.** *Under assumptions (2, 3), for any $0 < \alpha_t \leq \bar{\alpha}$, we have the following bound on the expected $\ell_2$ diameter of gradients:*

$$\mathbb{E}_{\vec{\xi}_t|\vec{\theta}_t} \Delta_2\left(\vec{g}_t\right)^2 \leq (1+\alpha_t)K^2 + 16\bar{\alpha}\alpha_t^{-1}\left(L^2\Delta_2(\vec{\theta}_t)^2 + h\sigma_t^2\right). \tag{24}$$

*Proof.* Note that we have

$$g_t^{(j)} = \nabla\mathcal{L}^{(j)}\left(\theta_t^{(j)}\right) + \xi_t^{(j)} \tag{25}$$

$$= \nabla\mathcal{L}^{(j)}\left(\bar{\theta}_t\right) + \left(\left(\nabla\mathcal{L}^{(j)}\left(\theta_t^{(j)}\right) - \nabla\mathcal{L}^{(j)}\left(\bar{\theta}_t\right)\right) + \xi_t^{(j)}\right). \tag{26}$$

Applying Lemma 1 with $\alpha = \alpha_t^{-1}$, and then Lemma 2 to the last terms then yields

$$\Delta_2\left(\vec{g}_t\right)^2 \leq (1+\alpha_t)\Delta_2\left(\overrightarrow{\nabla\mathcal{L}}\left(\bar{\theta}_t\right)\right)^2 \tag{27}$$

$$+ (1+\alpha_t^{-1})\left(2\Delta_2\left(\overrightarrow{\nabla\mathcal{L}}\left(\vec{\theta}_t\right) - \overrightarrow{\nabla\mathcal{L}}\left(\bar{\theta}_t\right)\right)^2 + 2\Delta_2\left(\vec{\xi}_t\right)^2\right). \tag{28}$$

Note that

$$\Delta_2\left(\overrightarrow{\nabla\mathcal{L}}\left(\bar{\theta}_t\right)\right)^2 = \max_{j,k\in[h]}\left\|\nabla\mathcal{L}^{(j)}\left(\bar{\theta}_t\right) - \nabla\mathcal{L}^{(k)}\left(\bar{\theta}_t\right)\right\|_2^2 \leq K^2. \tag{29}$$

The second term can be controlled using Lemma 3, which yields

$$\Delta_2\left(\overrightarrow{\nabla\mathcal{L}}\left(\vec{\theta}_t\right) - \overrightarrow{\nabla\mathcal{L}}\left(\bar{\theta}_t\right)\right) \leq 2\max_{j\in[h]}\left\|\nabla\mathcal{L}^{(j)}\left(\theta_t^{(j)}\right) - \nabla\mathcal{L}^{(j)}\left(\bar{\theta}_t\right)\right\|_2 \tag{30}$$

$$\leq 2\max_{j\in[h]} L\left\|\theta_t^{(j)} - \bar{\theta}_t\right\|_2 \leq 2L\Delta_2(\vec{\theta}_t). \tag{31}$$

To bound the third term, first note that Lemma 3 implies that $\Delta_2(\vec{\xi}_t) \leq 2\max_{j\in[h]}\left\|\xi_t^{(j)}\right\|_2$. Thus,

$$\mathbb{E}_{\vec{\xi}_t|\vec{\theta}_t} \Delta_2(\vec{\xi}_t)^2 \leq 4\mathbb{E}_{\vec{\xi}_t|\vec{\theta}_t} \max_{j\in[h]}\left\|\xi_t^{(j)}\right\|_2^2 \leq 4\mathbb{E}_{\vec{\xi}_t|\vec{\theta}_t} \sum_{j\in[h]}\left\|\xi_t^{(j)}\right\|_2^2 \tag{32}$$

$$= 4h\mathbb{E}_{\vec{\xi}_t|\vec{\theta}_t}\left\|\xi_t^{(j)}\right\|_2^2 \leq 4h\sigma_t^2. \tag{33}$$

Combining it all, and using $1 + \alpha_t^{-1} \leq (\bar{\alpha}+1)\alpha_t^{-1} \leq 2\bar{\alpha}\alpha_t^{-1}$ for $\alpha_t \leq \bar{\alpha}$ and $\bar{\alpha} = \max\{1, \sigma\} \geq 1$, yields the result. $\square$

**Definition 3.** *The (stochastic) effective gradient $G_t^{(j)}$ of node $j$ is defined by*

$$G_t^{(j)} = \frac{\theta_t^{(j)} - \theta_{t+1}^{(j)}}{\eta}. \tag{34}$$

In particular, we shall focus on the effective gradient of the average parameter, which turns out to also be the average of the effective gradients, that is,

$$\bar{G}_t \triangleq \frac{1}{h}\sum_{j\in[h]} G_t^{(j)} = \frac{\bar{\theta}_t - \bar{\theta}_{t+1}}{\eta}. \tag{35}$$

**Lemma 5.** *Under assumptions (2, 3), for any $0 < \alpha_t \leq \bar{\alpha}$, the expected discrepancy between the average effective gradient and the true gradient at the average parameter is bounded as follows:*

$$\mathop{\mathbb{E}}_{\vec{\xi}_t, \text{AVG}|\vec{\theta}_t} \left\| \bar{G}_t - \nabla \bar{\mathcal{L}} \left( \bar{\theta}_t \right) \right\|_2^2 \leq (1 + \alpha_t) C^2 \mathop{\mathbb{E}}_{\vec{\xi}_t|\vec{\theta}_t} \Delta_2 \left( \vec{g}_t \right)^2$$

$$+ 6 \bar{\alpha} \alpha_t^{-1} \left[ \frac{\sigma_t^2}{h} + \left( L^2 + \frac{2C^2}{\eta^2} \right) \Delta_2 \left( \vec{\theta}_t \right)^2 + \frac{2C^2}{t^2} \mathop{\mathbb{E}}_{\vec{\xi}_t|\vec{\theta}_t} \Delta_2 \left( \vec{g}_t \right)^2 \right]. \tag{36}$$

*Proof.* Note that

$$\bar{\theta}_{t+1} - \bar{\theta}_t = (\bar{\theta}_{t+1} - \bar{\theta}_{t+1/2}) + (\bar{\theta}_{t+1/2} - \bar{\theta}_t) \tag{37}$$

$$= (\bar{\theta}_{t+1} - \bar{\theta}_{t+1/2}) - \eta \bar{\gamma}_t. \tag{38}$$

As a result $\bar{G}_t = \bar{g}_t + (\bar{\gamma}_t - \bar{g}_t) - \frac{1}{\eta}(\bar{\theta}_{t+1} - \bar{\theta}_{t+1/2})$. Moreover, we have

$$\bar{g}_t = \frac{1}{h} \sum_{j \in [h]} \nabla \mathcal{L}^{(j)} \left( \theta_t^{(j)} \right) + \frac{1}{h} \sum_{j \in [h]} \xi_t^{(j)} \tag{39}$$

$$= \nabla \bar{\mathcal{L}} \left( \bar{\theta}_t \right) + \frac{1}{h} \sum_{j \in [h]} \left( \nabla \mathcal{L}^{(j)} \left( \theta_t^{(j)} \right) - \nabla \mathcal{L}^{(j)} \left( \bar{\theta}_t \right) \right) + \frac{1}{h} \sum_{j \in [h]} \xi_t^{(j)}, \tag{40}$$

where $\nabla \bar{\mathcal{L}} \left( \bar{\theta}_t \right) = \frac{1}{h} \sum_{j \in [h]} \nabla \mathcal{L}^{(j)} \left( \bar{\theta}_t \right)$ is the average gradient at the average parameter. This then yields:

$$\bar{G}_t - \nabla \bar{\mathcal{L}} \left( \bar{\theta}_t \right) = \frac{1}{h} \sum_{j \in [h]} \left( \nabla \mathcal{L}^{(j)} \left( \theta_t^{(j)} \right) - \nabla \mathcal{L}^{(j)} \left( \bar{\theta}_t \right) \right) + \frac{1}{h} \sum_{j \in [h]} \xi_t^{(j)}$$

$$+ \frac{1}{\eta} \left( \bar{\theta}_{t+1/2} - \bar{\theta}_{t+1} \right) + \left( \bar{\gamma}_t - \bar{g}_t \right). \tag{41}$$

Applying Lemma 1 for $\alpha = \alpha_t$ (by isolating the first three terms), and then Lemma 2 to the first three terms then yields

$$\left\| \bar{G}_t - \nabla \bar{\mathcal{L}} \left( \bar{\theta}_t \right) \right\|_2^2 \leq 3(1 + \alpha_t^{-1}) \left\| \frac{1}{h} \sum_{j \in [h]} \left( \nabla \mathcal{L}^{(j)} \left( \theta_t^{(j)} \right) - \nabla \mathcal{L}^{(j)} \left( \bar{\theta}_t \right) \right) \right\|_2^2$$

$$+ 3(1 + \alpha_t^{-1}) \left\| \frac{1}{h} \sum_{j \in [h]} \xi_t^{(j)} \right\|_2^2 + \frac{3(1 + \alpha_t^{-1})}{\eta^2} \left\| \bar{\theta}_{t+1/2} - \bar{\theta}_{t+1} \right\|_2^2$$

$$+ (1 + \alpha_t) \left\| \bar{\gamma}_t - \bar{g}_t \right\|_2^2. \tag{42}$$

We now note that the expectation of each term can be bounded. Indeed,

$$\left\| \frac{1}{h} \sum_{j \in [h]} \left( \nabla \mathcal{L}^{(j)} \left( \theta_t^{(j)} \right) - \nabla \mathcal{L}^{(j)} \left( \bar{\theta}_t \right) \right) \right\|_2 \leq \frac{1}{h} \sum_{j \in [h]} \left\| \nabla \mathcal{L}^{(j)} \left( \theta_t^{(j)} \right) - \nabla \mathcal{L}^{(j)} \left( \bar{\theta}_t \right) \right\|_2 \tag{43}$$

$$\leq \frac{1}{h} \sum_{j \in [h]} L \left\| \theta_t^{(j)} - \bar{\theta}_t \right\|_2 \leq \frac{1}{h} \sum_{j \in [h]} L \Delta_2(\vec{\theta}_t) = L \Delta_2(\vec{\theta}_t). \tag{44}$$

Moreover, using the conditional non-correlation of $\xi_t^{(j)}$, we have

$$\mathop{\mathbb{E}}_{\xi_t|\vec{\theta}_t} \left\| \frac{1}{h} \sum_{j \in [h]} \xi_t^{(j)} \right\|_2^2 = \mathop{\mathbb{E}}_{\xi_t|\vec{\theta}_t} \frac{1}{h^2} \sum_{j,k \in [h]} \xi_t^{(j)} \cdot \xi_t^{(k)} = \frac{1}{h^2} \sum_{j,k \in [h]} \mathop{\mathbb{E}}_{\xi_t|\vec{\theta}_t} \xi_t^{(j)} \cdot \xi_t^{(k)} \tag{45}$$

$$= \frac{1}{h^2} \sum_{j \in [h]} \mathop{\mathbb{E}}_{\xi_t|\vec{\theta}_t} \left\| \xi_t^{(j)} \right\|_2^2 \leq \frac{1}{h^2} \sum_{j \in [h]} \sigma_t^2 = \frac{\sigma_t^2}{h}. \tag{46}$$

For the third term, we use the $C$-averaging guarantee of AVG to obtain

$$\mathop{\mathbb{E}}_{\text{AVG}|\bar{\theta}_{t+1/2}} \left\| \bar{\theta}_{t+1} - \bar{\theta}_{t+1/2} \right\|_2^2 \leq C^2 \Delta_2(\vec{\theta}_{t+1/2})^2. \tag{47}$$

Since $\vec{\theta}_{t+1/2} = \vec{\theta}_t - \eta\vec{\gamma}_t$, using Lemma 2 and taking the expectation over $\vec{\xi}_t$, we then have

$$\mathop{\mathbb{E}}_{\vec{\xi}_t, \text{AVG}|\vec{\theta}_t} \left\| \bar{\theta}_{t+1} - \bar{\theta}_{t+1/2} \right\|_2^2 \leq 2C^2 \Delta_2(\vec{\theta}_t)^2 + 2C^2\eta^2 \mathop{\mathbb{E}}_{\vec{\xi}_t, \text{AVG}|\vec{\theta}_t} \Delta_2(\vec{\gamma}_t)^2 \tag{48}$$

$$\leq 2C^2 \Delta_2(\vec{\theta}_t)^2 + 2C^2\eta^2 \frac{\mathbb{E}_{\vec{\xi}_t|\vec{\theta}_t} \Delta_2(\vec{g}_t)^2}{t^2} \tag{49}$$

Finally, for the last term, we again use the $C$-averaging guarantee of the aggregation AVG and take the expectation over $\vec{\xi}_t$:

$$\mathop{\mathbb{E}}_{\vec{\xi}_t, \text{AVG}|\vec{\theta}_t} \left\| \bar{\gamma}_t - \bar{g}_t \right\|_2^2 \leq C^2 \mathop{\mathbb{E}}_{\vec{\xi}_t|\vec{\theta}_t} \Delta_2(\vec{g}_t)^2. \tag{50}$$

Combining it all, and using $1 + \alpha_t^{-1} \leq 2\bar{\alpha}\alpha_t^{-1}$ finally yields the lemma. $\qquad\square$

**Lemma 6.** *Under assumptions (2, 3), for any $0 < \alpha_t \leq \bar{\alpha}$ and $\alpha_t \geq 1/\sqrt{t}$, there exist constants $A$ and $B$ which can be computed explicitly given $\bar{\alpha}$, $C$, $L$, $h$ and $\eta$, such that*

$$\mathop{\mathbb{E}}_{\vec{\xi}_t, \text{AVG}|\vec{\theta}_t} \left\| \bar{G}_t - \nabla\mathcal{L}\left(\bar{\theta}_t\right) \right\|_2^2 \leq (1 + \alpha_t)^2 (1 + \kappa_t) C^2 K^2 + \alpha_t^{-1}\left(A\Delta_2(\vec{\theta}_t)^2 + B\sigma_t^2\right), \tag{51}$$

*where $\kappa_t \leq \frac{12\bar{\alpha}}{t^{(3/2)}}$.*

*Proof.* Combining the two previous lemmas, this bound can be guaranteed by setting

$$\kappa_t = \frac{12\bar{\alpha}}{(1 + \alpha_t)\alpha_t t^2} \tag{52}$$

$$A_t = 16(1 + \alpha_t)\bar{\alpha}C^2 L^2 + 6\bar{\alpha}L^2 + \frac{12\bar{\alpha}C^2}{\eta^2} + \frac{192\bar{\alpha}^2 C^2 L^2}{\alpha_t t^2} \tag{53}$$

$$B_t = 16(1 + \alpha_t)\bar{\alpha}C^2 h + \frac{6\bar{\alpha}}{h} + \frac{192\bar{\alpha}^2 C^2 h}{\alpha_t t^2}. \tag{54}$$

Assumptions $0 < \alpha_t \leq \bar{\alpha}$ (which implies $1 + \alpha_t \leq 2\bar{\alpha}$) and $\alpha_t\sqrt{t} \geq 1$ allow to conclude, with

$$\kappa_t \leq \frac{12\bar{\alpha}}{t^{(3/2)}} \tag{55}$$

$$A = 32\bar{\alpha}^2 C^2 L^2 + 6\bar{\alpha}L^2 + \frac{12\bar{\alpha}C^2}{\eta^2} + 192\bar{\alpha}^2 C^2 L^2 \tag{56}$$

$$B = 32\bar{\alpha}^2 C^2 h + \frac{6\bar{\alpha}}{h} + 192\bar{\alpha}^2 C^2 h. \tag{57}$$

This shows in particular that $A$ and $B$ can indeed be computed from the different constants of the problem. $\qquad\square$

**Lemma 7.** *We have the following bound on parameter drift:*

$$\mathop{\mathbb{E}}_{\text{AVG}|\vec{\theta}_t, \vec{g}_t} \Delta_2(\vec{\theta}_{t+1})^2 \leq \frac{1}{2}\Delta_2(\vec{\theta}_t)^2 + \frac{\eta^2}{2t^2}\Delta_2(\vec{g}_t)^2. \tag{58}$$

*Proof.* Recall that $\vec{\theta}_{t+1} = \overrightarrow{\text{AVG}_1} \circ \overrightarrow{\text{BYZ}}_{t,\theta}(\vec{\theta}_{t+1/2})$. Thus, by the asymptotic agreement property of $\text{AVG}_1$, we know that

$$\mathop{\mathbb{E}}_{\text{AVG}|\vec{\theta}_t, \vec{g}_t} \Delta_2(\vec{\theta}_{t+1})^2 \leq \frac{1}{4}\Delta_2(\vec{\theta}_{t+1/2})^2. \tag{59}$$

Now recall that $\vec{\theta}_{t+1/2} = \vec{\theta}_t - \eta\vec{\gamma}_t$. Applying Lemma 1 for $\alpha = 1$ thus yields

$$\Delta_2(\vec{\theta}_{t+1/2})^2 \leq 2\Delta_2(\vec{\theta}_t)^2 + 2\eta^2\Delta_2(\vec{\gamma}_t)^2. \tag{60}$$

We now use the asymptotic agreement property of $\text{AVG}_{N(t)}$, which yields

$$\mathbb{E}_{\text{AVG}|\vec{\theta}_t,\vec{g}_t} \Delta_2(\vec{\gamma}_t)^2 \leq \frac{\Delta_2(\vec{g}_t)^2}{t^2}. \tag{61}$$

Combining it all then yields the result. $\qquad\square$

**Lemma 8.** *Under assumptions (2, 3), $0 < \alpha_t \leq \bar{\alpha}$ and $\alpha_t = \max\{1/\sqrt{t}, \sigma_t\}$, there exists a constant $D$ such that*

$$\mathbb{E}_{\vec{\xi}_{1:t}} \Delta_2(\vec{\theta}_t)^2 \leq D/t^2. \tag{62}$$

*Note that the constant $D$ can be computed from the constants $L$, $\eta$, $K$, $h$ and the functions $\sigma_t$ and $\alpha_t$.*

*Proof.* Denote $u_t = \mathbb{E}_{\vec{\xi}_{1:t}} \Delta_2(\vec{\theta}_t)^2$. Combining Lemmas 4 and 7 yields

$$u_{t+1} \leq \rho_t u_t + \delta_t, \tag{63}$$

where $\rho_t$ and $\delta_t$ are given by

$$\rho_t \triangleq \frac{1}{2} + \frac{8\bar{\alpha}L^2\eta^2}{\alpha_t t^2} \tag{64}$$

$$\delta_t \triangleq \frac{\eta^2}{2t^2}\left((1+\alpha_t)K^2 + 16\bar{\alpha}\alpha_t^{-1}h\sigma_t^2\right). \tag{65}$$

Given that $\alpha_t \geq 1/\sqrt{t}$, we know that $\rho_t \leq \frac{1}{2} + 8\bar{\alpha}L^2\eta^2 t^{-3/2}$. Thus, for $t \geq t_0 \triangleq \left(32\bar{\alpha}L^2\eta^2\right)^{2/3}$, we know that $\rho_t \leq \rho \triangleq 3/4$. Moreover, using now $\sigma_t \leq \alpha_t \leq \bar{\alpha}$, we know that

$$\delta_t \leq \frac{\eta^2}{2t^2}\left((1+\bar{\alpha})K^2 + 16\bar{\alpha}h\sigma_t\right) \leq \frac{\eta^2}{2t^2}\left((1+\bar{\alpha})K^2 + 16h\bar{\alpha}^2\right) \triangleq \delta_t^+. \tag{66}$$

Now note that $\delta_t^+$ is decreasing. In particular, for $t \geq t_0$ we now have

$$u_{t+1} \leq \rho u_t + \delta_t^+. \tag{67}$$

By induction we see that, for $t \geq 1$, we have

$$u_{t+t_0} \leq \rho^t u_{t_0} + \sum_{s=0}^{t-1} \rho^s \delta_{t+t_0-s-1}^+. \tag{68}$$

We now separate the sum into two parts. Calling $t_1$ the separation point for $t_1 \geq 1$, and using the fact that $\delta_t^+$ is decreasing yields

$$u_{t+t_0} \leq \rho^t u_{t_0} + \sum_{s=0}^{t_1-1} \rho^s \delta_{t+t_0-s-1}^+ + \sum_{s=t_1}^{t-1} \rho^s \delta_{t+t_0-s-1}^+ \tag{69}$$

$$\leq \rho^t u_{t_0} + \delta_{t+t_0-t_1}^+ \sum_{s=0}^{t_1-1} \rho^s + \delta_{t_0}^+ \sum_{s=t_1}^{t-1} \rho^s \tag{70}$$

$$\leq \rho^t u_{t_0} + \delta_{t+t_0-t_1}^+ \sum_{s=0}^{\infty} \rho^s + \delta_{t_0}^+ \sum_{s=t_1}^{\infty} \rho^s \tag{71}$$

$$\leq \rho^t u_{t_0} + \frac{\delta_{t+t_0-t_1}^+}{1-\rho} + \frac{\rho^{t_1}\delta_{t_0}^+}{1-\rho} \tag{72}$$

$$= \rho^t u_{t_0} + 4\delta_{t+t_0-t_1}^+ + 4\rho^{t_1}\delta_{t_0}^+. \tag{73}$$

We now take $t_1 = \left\lfloor \frac{t+t_0}{2} \right\rfloor$. As a result,

$$\delta_{t+t_0-t_1}^+ = \delta_{\lceil \frac{t+t_0}{2} \rceil}^+ \leq \frac{2\eta^2}{(t+t_0)^2}\left((1+\bar{\alpha})K^2 + 16h\bar{\alpha}^2\right). \tag{74}$$

Now define $v_t$ by $v_0 = 0$ and $v_{t+1} = \rho_t v_t + \delta_t$. Note that $u_{t_0}$ can be upper-bounded given $L$, $\eta$, $\alpha_{1:t_0}$, $K$, $h$ and $\sigma_{1:t}$, by computing $v_{t_0}$. Indeed, by induction we then clearly have $u_{t_0} \le v_{t_0}$, and thus the bound

$$u_{t+t_0} \le \frac{8\eta^2 \left((1+\bar{\alpha})K^2 + 16h\bar{\alpha}^2\right)}{(t+t_0)^2} + \rho^t v_{t_0} + 4\rho^{t_1} \delta_{t_0}^+, \tag{75}$$

where the right-hand side is perfectly computable given the constants of the problem. Given that $\rho^{t_1} = \mathcal{O}(1/t^2)$ and $\rho^t = \mathcal{O}(1/t^2)$, we can also compute a constant $D$ from these constants, such that for all iterations $t$, we have $u_t \le D/t^2$. $\qquad\square$

## A.1 Reduction from collaborative learning to averaging agreement

Now we proceed with the proof of our theorem.

*Proof of Theorem 1.* At any iteration $t$, Taylor's theorem implies the existence of $\lambda \in [0,1]$ such that

$$\bar{\mathcal{L}}\left(\bar{\theta}_{t+1}\right) = \bar{\mathcal{L}}\left(\bar{\theta}_t - \eta\bar{G}_t\right) \tag{76}$$

$$= \bar{\mathcal{L}}\left(\bar{\theta}_t\right) - \eta\bar{G}_t \cdot \nabla\bar{\mathcal{L}}\left(\bar{\theta}_t\right) + \frac{1}{2}\left(\eta\bar{G}_t\right)^T \nabla^2\bar{\mathcal{L}}\left(\bar{\theta}_t - \lambda\eta\bar{G}_t\right)\left(\eta\bar{G}_t\right). \tag{77}$$

Lipschitz continuity of the gradient implies that $\nabla^2\bar{\mathcal{L}}\left(\bar{\theta}_t - \lambda\eta\bar{G}_t\right) \preceq LI$, which thus implies

$$\bar{\mathcal{L}}\left(\bar{\theta}_{t+1}\right) \le \bar{\mathcal{L}}\left(\bar{\theta}_t\right) - \eta\bar{G}_t \cdot \nabla\bar{\mathcal{L}}\left(\bar{\theta}_t\right) + \frac{L\eta^2}{2}\left\|\bar{G}_t\right\|_2^2. \tag{78}$$

For the second term, using the inequality $2u \cdot v \ge -\|u\|_2^2 - \|v\|_2^2$, note that

$$\bar{G}_t \cdot \nabla\bar{\mathcal{L}}\left(\bar{\theta}_t\right) = \left(\bar{G}_t - \nabla\bar{\mathcal{L}}\left(\bar{\theta}_t\right) + \nabla\bar{\mathcal{L}}\left(\bar{\theta}_t\right)\right) \cdot \nabla\bar{\mathcal{L}}\left(\bar{\theta}_t\right) \tag{79}$$

$$= \left(\bar{G}_t - \nabla\bar{\mathcal{L}}\left(\bar{\theta}_t\right)\right) \cdot \nabla\bar{\mathcal{L}}\left(\bar{\theta}_t\right) + \left\|\nabla\bar{\mathcal{L}}\left(\bar{\theta}_t\right)\right\|_2^2 \tag{80}$$

$$\ge -\frac{1}{2}\left\|\bar{G}_t - \nabla\bar{\mathcal{L}}\left(\bar{\theta}_t\right)\right\|_2^2 - \frac{1}{2}\left\|\nabla\bar{\mathcal{L}}\left(\bar{\theta}_t\right)\right\|_2^2 + \left\|\nabla\bar{\mathcal{L}}\left(\bar{\theta}_t\right)\right\|_2^2 \tag{81}$$

$$= -\frac{1}{2}\left\|\bar{G}_t - \nabla\bar{\mathcal{L}}\left(\bar{\theta}_t\right)\right\|_2^2 + \frac{1}{2}\left\|\nabla\bar{\mathcal{L}}\left(\bar{\theta}_t\right)\right\|_2^2. \tag{82}$$

For the last term, we use $\|a+b\|_2^2 \le 2\|a\|_2^2 + 2\|b\|_2^2$ to derive

$$\left\|\bar{G}_t\right\|_2^2 = \left\|\bar{G}_t - \nabla\bar{\mathcal{L}}\left(\bar{\theta}_t\right) + \nabla\bar{\mathcal{L}}\left(\bar{\theta}_t\right)\right\|_2^2 \tag{83}$$

$$\le 2\left\|\bar{G}_t - \nabla\bar{\mathcal{L}}\left(\bar{\theta}_t\right)\right\|_2^2 + 2\left\|\nabla\bar{\mathcal{L}}\left(\bar{\theta}_t\right)\right\|_2^2. \tag{84}$$

Combining the two above bounds into Equation (78) yields

$$\bar{\mathcal{L}}\left(\bar{\theta}_{t+1}\right) \le \bar{\mathcal{L}}\left(\bar{\theta}_t\right) - \left(\frac{\eta}{2} - L\eta^2\right)\left\|\nabla\bar{\mathcal{L}}\left(\bar{\theta}_t\right)\right\|_2^2 + \left(\frac{\eta}{2} + L\eta^2\right)\left\|\bar{G}_t - \nabla\bar{\mathcal{L}}\left(\bar{\theta}_t\right)\right\|_2^2. \tag{85}$$

Rearranging the terms then yields

$$\left(\frac{\eta}{2} - L\eta^2\right)\left\|\nabla\bar{\mathcal{L}}\left(\bar{\theta}_t\right)\right\|_2^2 \le \bar{\mathcal{L}}\left(\bar{\theta}_t\right) - \bar{\mathcal{L}}\left(\bar{\theta}_{t+1}\right) + \left(\frac{\eta}{2} + L\eta^2\right)\left\|\bar{G}_t - \nabla\bar{\mathcal{L}}\left(\bar{\theta}_t\right)\right\|_2^2. \tag{86}$$

We now use the fact that $\eta \le \delta/12L$. Denoting $\nu \triangleq \delta/6$, this implies that $\frac{\eta}{2} - L\eta^2 \ge (1-\nu)\eta/2$ and $\frac{\eta}{2} + L\eta^2 \le (1+\nu)\eta/2$. As a result,

$$\left\|\nabla\bar{\mathcal{L}}\left(\bar{\theta}_t\right)\right\|_2^2 \le \frac{4}{\eta}\left(\bar{\mathcal{L}}\left(\bar{\theta}_t\right) - \bar{\mathcal{L}}\left(\bar{\theta}_{t+1}\right)\right) + \frac{1+\nu}{1-\nu}\left\|\bar{G}_t - \nabla\bar{\mathcal{L}}\left(\bar{\theta}_t\right)\right\|_2^2. \tag{87}$$

Taking the expectation and the average over $t \in [T]$ yields

$$\mathbb{E}_{\vec{\xi}_{1:T}} \frac{1}{T}\sum_{t\in[T]}\left\|\nabla\bar{\mathcal{L}}\left(\bar{\theta}_t\right)\right\|_2^2 \le \frac{4(\bar{\mathcal{L}}\left(\bar{\theta}_1\right) - \bar{\mathcal{L}}\left(\bar{\theta}_{T+1}\right))}{\eta T} + \frac{1+\nu}{1-\nu}\frac{1}{T}\sum_{t\in[T]}\mathbb{E}_{\vec{\xi}_{1:T}}\left\|\bar{G}_t - \nabla\bar{\mathcal{L}}\left(\bar{\theta}_t\right)\right\|_2^2. \tag{88}$$

Note that $\mathbb{E}_{\vec{\xi}_{1:T}} \frac{1}{T} \sum_{t \in [T]} \left\| \nabla \bar{\mathcal{L}} \left( \bar{\theta}_t \right) \right\|_2^2 = \mathbb{E} \left\| \nabla \bar{\mathcal{L}} \left( \bar{\theta}_* \right) \right\|_2^2$, since the second term is obtained by taking uniformly randomly one of the values averaged in the first term. Using also the fact that $\bar{\mathcal{L}} \left( \bar{\theta}_{T+1} \right) \geq \inf_\theta \bar{\mathcal{L}} (\theta) \geq 0$ (Assumption 1) and $\mathcal{L}^{(j)}(\theta_1) \leq \mathcal{L}_{max}$, we then obtain

$$\mathbb{E} \left\| \nabla \bar{\mathcal{L}} \left( \bar{\theta}_* \right) \right\|_2^2 \leq \frac{4 \mathcal{L}_{max}}{\eta T} + \frac{1 + \nu}{1 - \nu} \frac{1}{T} \sum_{t \in [T]} \mathbb{E}_{\vec{\xi}_{1:T}} \left\| \bar{G}_t - \nabla \bar{\mathcal{L}} \left( \bar{\theta}_t \right) \right\|_2^2. \tag{89}$$

Now recall that all nodes started with the same value $\theta_1$, and thus know $\bar{\theta}_1$. As a result, each node $j$ can compute $\mathcal{L}^{(j)}(\bar{\theta}_1)$, but it cannot compute $\bar{\mathcal{L}} \left( \bar{\theta}_1 \right)$.

Let us focus on the last term. Taking Lemma 6 and averaging over all noises $\vec{\xi}_{1:t}$ yields

$$\mathbb{E}_{\vec{\xi}_{1:t}} \left\| \bar{G}_t - \nabla \bar{\mathcal{L}} \left( \bar{\theta}_t \right) \right\|_2^2 \leq (1 + \alpha_t)^2 (1 + \kappa_t) C^2 K^2 + \alpha_t^{-1} \left( A \, \mathbb{E}_{\vec{\xi}_{1:t}} \Delta_2(\vec{\theta}_t)^2 + B \sigma_t^2 \right). \tag{90}$$

Recall that, by Lemma 8, $\mathbb{E}_{\vec{\xi}_{1:t}} \Delta_2(\vec{\theta}_t)^2 \leq D/t^2$. Recall also that $\alpha_t \triangleq \max \left\{ 1/\sqrt{t}, \sigma_t \right\}$ and $\kappa_t \leq 12 \bar{\alpha}/t^{(3/2)}$. Then we obtain

$$\mathbb{E}_{\vec{\xi}_{1:t}} \left\| \bar{G}_t - \nabla \bar{\mathcal{L}} \left( \bar{\theta}_t \right) \right\|_2^2 \leq (1 + \alpha_t)^2 \left( 1 + \frac{12 \bar{\alpha}}{t^{3/2}} \right) C^2 K^2 + \frac{AD}{t^{3/2}} + B \sigma_t. \tag{91}$$

Now recall that we regularly increase the batch size (see Section 2.2). Thus, there exists some iteration $T_1$, such that $\sigma_{T_1} \leq \min \left\{ \nu, \nu C^2 K^2/8B \right\} = \mathcal{O}(\delta)$. Defining $T_2 \triangleq 1/\nu^2$, $T_3 \triangleq (12 \bar{\alpha}/\nu)^{2/3}$ and $T_4 \triangleq (8AD/\nu C^2 K^2)^{2/3}$, for $t \geq T_5 \triangleq \max \left\{ T_1, T_2, T_3, T_4 \right\}$, we have

$$\mathbb{E}_{\vec{\xi}_{1:t}} \left\| \bar{G}_t - \nabla \bar{\mathcal{L}} \left( \bar{\theta}_t \right) \right\|_2^2 \leq (1 + \nu)^3 C^2 K^2 + \nu C^2 K^2/8 + \nu C^2 K^2/8 \tag{92}$$

$$\leq (1 + 5\nu) C^2 K^2, \tag{93}$$

using the inequality $\nu \leq 1/2$ to show that $(1 + \nu)^3 \leq 1 + 3\nu + 3\nu^2 + \nu^3 \leq 1 + 3\nu + 3\nu/2 + \nu/4$. If we now average this quantity over $t$ from 1 to $T$, assuming $T \geq T_5$, we can separate the sum from 1 to $T_5$, and the sum from $T_5 + 1$ to $T$. This yields

$$\frac{1}{T} \sum_{t \in T} \mathbb{E}_{\vec{\xi}_{1:t}} \left\| \bar{G}_t - \nabla \bar{\mathcal{L}} \left( \bar{\theta}_t \right) \right\|_2^2 \leq \frac{T_5 E}{T} + \frac{T - T_5}{T} (1 + 5\nu) C^2 K^2, \tag{94}$$

where $E = (1 + \bar{\alpha})^2 (1 + 12 \bar{\alpha}) C^2 K^2 + AD + B\sigma$. Now consider $T_6 \triangleq T_5 E / \nu C^2 K^2$. For $T \geq T_6$, we then have

$$\frac{1}{T} \sum_{t \in T} \mathbb{E}_{\vec{\xi}_{1:t}} \left\| \bar{G}_t - \nabla \bar{\mathcal{L}} \left( \bar{\theta}_t \right) \right\|_2^2 \leq (1 + 6\nu) C^2 K^2. \tag{95}$$

Plugging this into Equation (89), and using $1/(1 - \nu) \leq 1 + 2\nu$ for $0 < \nu \leq 1/2$ then yields, for $T \geq T_6$,

$$\mathbb{E} \left\| \nabla \bar{\mathcal{L}} \left( \bar{\theta}_* \right) \right\|_2^2 \leq \frac{4 \mathcal{L}_{max}}{\eta T} + (1 + \nu)(1 + 2\nu)(1 + 6\nu) C^2 K^2. \tag{96}$$

Now note that $(1 + \nu)(1 + 2\nu) \leq 1 + 4\nu$ for $\nu \leq 1/2$. Now consider $T_7 \triangleq \mathcal{L}_{max}/(\nu^2 \eta C^2 K^2)$. Then for $T \geq T_8 \triangleq \max \left\{ T_6, T_7 \right\}$, we have the guarantee

$$\mathbb{E} \left\| \nabla \bar{\mathcal{L}} \left( \bar{\theta}_* \right) \right\|_2^2 \leq \left( (1 + 4\nu)(1 + 6\nu) + \nu^2) \right) C^2 K^2 \leq (1 + 6\nu)^2 C^2 K^2. \tag{97}$$

Now recall that $\nu = \delta/6$, and consider $T_9 \triangleq D\tau^2/24\delta^2$ and[8] $T = T_{\text{LEARN}}(\delta) \triangleq \max \left\{ T_8, T_9 \right\}$. Note that we have

$$\mathbb{E} \Delta_2 \left( \vec{\theta}_* \right)^2 = \frac{1}{T} \sum_{t \in [T]} \mathbb{E} \Delta_2 \left( \vec{\theta}_t \right)^2 \leq \frac{1}{T} \sum_{t \in [T]} \frac{D}{t^2} \tag{98}$$

$$\leq \frac{D}{T} \sum_{t=1}^\infty \frac{1}{t^2} = \frac{D}{T} \frac{\tau^2}{24} \leq \delta^2. \tag{99}$$

Combining it all yields

$$\mathbb{E} \Delta_2 \left( \vec{\theta}_* \right)^2 \leq \delta^2 \quad \text{and} \quad \mathbb{E} \left\| \nabla \bar{\mathcal{L}} \left( \bar{\theta}_* \right) \right\|_2^2 \leq (1 + \delta)^2 C^2 K^2, \tag{100}$$

which corresponds to saying that LEARN solves collaborative learning. $\qquad\square$

---

[8]Here, $\tau \approx 6.2832$ is the ratio of the circumference of the circle by its radius.

## A.2 Reduction from averaging agreement to collaborative learning

*Proof of Theorem 2.* Without loss of generality, assume $0 < \delta \leq 1$. Let $\vec{x} \in \mathbb{R}^{d \cdot h}$ be a family of vectors. For any honest node $j \in [h]$, consider the losses defined by $\mathcal{L}^{(j)}(\theta) \triangleq \frac{1}{2} \left\| \theta - x^{(j)} \right\|_2^2$. Note that we thus have $\nabla \mathcal{L}^{(j)}(\theta) = \theta - x^{(j)}$. As a result,

$$\left\| \nabla \mathcal{L}^{(j)}(\theta) - \nabla \mathcal{L}^{(k)}(\theta) \right\|_2 = \left\| x^{(j)} - x^{(k)} \right\|_2 \leq \Delta_2(\vec{x}), \tag{101}$$

which corresponds to saying that local losses satisfy the definition of collaborative learning (Definition 1) with $K = \Delta_2(\vec{x})$. Now consider a Byzantine-resilient $C$-collaborative learning algorithm LEARN. For any $N \in \mathbb{N}$, we run LEARN with parameter $\delta \triangleq \min \left\{ 1, \Delta_2(\vec{x})/2^N \right\}$, which outputs $\vec{x}_N \triangleq \vec{\theta}_*$.

We then have the guarantee $\mathbb{E} \, \Delta_2(\vec{x}_N)^2 = \mathbb{E} \, \Delta_2(\vec{\theta}_*)^2 \leq \delta^2 \leq \Delta_2(\vec{x})^2 / 4^N$, which corresponds to asymptotic agreement. Moreover, we notice that

$$\nabla \bar{\mathcal{L}}(\bar{\theta}_*) = \frac{1}{h} \sum_{j \in [h]} (\bar{\theta}_* - x^{(j)}) = \bar{\theta}_* - \bar{x} = \bar{x}_N - \bar{x}. \tag{102}$$

The second collaborative learning guarantee of algorithm LEARN (Equation (4) in the paper) then yields

$$\mathbb{E} \left\| \bar{x}_N - \bar{x} \right\|_2^2 = \mathbb{E} \left\| \nabla \bar{\mathcal{L}}(\bar{\theta}_*) \right\|_2^2 \leq (1+\delta)^2 C^2 K^2 = \left( (1+\delta) C \right)^2 \Delta_2(\vec{x})^2. \tag{103}$$

This shows $(1+\delta)C$-averaging, and concludes the proof. $\qquad\square$

## B  Efficient i.i.d. algorithm

From a practical viewpoint, a disadvantage of Algorithm 1 is that it requires a large number of communication rounds due to the use of two instances of the averaging algorithm AVG, one of which requires a stronger agreement (which induces more communication rounds) as the iteration number $t$ grows. This is because when the data distributions of nodes vary from each other, data heterogeneity enables the Byzantines to bias the models in their favor and induce model drift more easily.

In this section, we show that we can perform better in the homogeneous (i.i.d.) setting, where the ability of Byzantines is more limited. More specifically, we present a simpler algorithm (Algorithm 2) that uses the averaging algorithm AVG only once at each iteration with a fixed parameter ($N = 1$). This results in much lower communication/computation overhead compared to Algorithm 1. We evaluate the throughput overhead of our two algorithms compared to the non-robust vanilla implantation in Section 5, which, not surprisingly, shows the superiority of the i.i.d. algorithm.

Note that in the homogeneous setting, all local losses are equal, i.e., $\mathcal{L}^{(j)}(\cdot) = \mathcal{L}(\cdot)$ for all honest nodes $j \in [h]$.

---

**Data:** Global loss gradient oracle
**Result:** Model parameters $\theta_t$

1  Initialize local parameters $\theta_1$ using a fixed seed $s$;
2  Fix learning rate $\eta \leq \frac{1}{2L}$;
3  Fix number of rounds $T \triangleq T_{\text{HOM-LEARN}}(\delta)$;
4  **for** $t \leftarrow 1, \ldots, T$ **do**
5     $g_t \leftarrow \texttt{GradientOracle}(\theta_t)$;
6     $\theta_{t+1/2} \leftarrow \theta_t - \eta g_t$;
7     $\theta_{t+1} \leftarrow \text{AVG}_1 \circ \text{BYZ}_{t,\theta}(\vec{\theta}_{t+1/2})$   // Vulnerable to Byzantine attacks
8  **end**
9  Draw $* \sim \mathcal{U}([T])$;
10  Return $\theta_*$;

**Algorithm 2:** HOM-LEARN execution on an honest node.

---

## B.1 Proof of Theorem 3

Before proving our theorem, we prove some preliminary lemmas.

**Lemma 9.** *Under Assumption 3, we have*

$$\mathbb{E}_{\vec{\xi}_t | \vec{\theta}_t} \Delta_2(\vec{\xi}_t)^2 \le 4\sigma_t^2 h, \tag{104}$$

*where $\xi_t^{(j)}$ is the gradient estimation error of node $j$ at round $t$ defined in (23).*

*Proof.*

$$\mathbb{E}_{\vec{\xi}_t | \vec{\theta}_t} \Delta_2(\vec{\xi}_t)^2 = \mathbb{E}_{\vec{\xi}_t | \vec{\theta}_t} \max_{j,k \in [h]} \left\| \xi_t^{(j)} - \xi_t^{(k)} \right\|_2^2 \tag{105}$$

$$\le 4 \mathbb{E}_{\vec{\xi}_t | \vec{\theta}_t} \max_{j \in [h]} \left\| \xi_t^{(j)} \right\|_2^2, \tag{106}$$

where we used Lemma 3. We now use the fact that the maximum over nodes $j \in [h]$ is smaller than the sum over nodes $j \in [h]$, yielding

$$\mathbb{E}_{\vec{\xi}_t | \vec{\theta}_t} \Delta_2(\vec{\xi}_t)^2 \le 4 \mathbb{E}_{\vec{\xi}_t | \vec{\theta}_t} \sum_{j \in [h]} \left\| \xi_t^{(j)} \right\|_2^2 = 4 \sum_{j \in [h]} \mathbb{E}_{\vec{\xi}_t | \vec{\theta}_t} \left\| \xi_t^{(j)} \right\|_2^2 \tag{107}$$

$$\le 4 \sum_{j \in [h]} \sigma_t^2 = 4\sigma_t^2 h, \tag{108}$$

where the last inequality uses Assumption 3. □

**Lemma 10.** *Under assumptions (2,3), the expected $\ell_2$ diameter between honest gradient estimations is upper-bounded as follows*

$$\mathbb{E}_{\vec{\xi}_t | \vec{\theta}_t} \Delta_2(\vec{g}_t)^2 \le 2L^2 \Delta_2\left(\vec{\theta}_t\right)^2 + 8\sigma_t^2 h. \tag{109}$$

*Proof.* By Lemma 2, we know that

$$\Delta_2(\vec{g}_t)^2 \le 2\Delta_2\left(\nabla\mathcal{L}\left(\vec{\theta}_t\right)\right)^2 + 2\Delta_2\left(\vec{\xi}_t\right)^2. \tag{110}$$

Assumption 2 then guarantees that

$$\Delta_2\left(\nabla\mathcal{L}\left(\vec{\theta}_t\right)\right)^2 = \max_{j,k \in [h]} \left\| \nabla\mathcal{L}\left(\theta_t^{(j)}\right) - \nabla\mathcal{L}\left(\theta_t^{(k)}\right) \right\|_2^2 \tag{111}$$

$$\le \max_{j,k \in [h]} L^2 \left\| \theta_t^{(j)} - \theta_t^{(k)} \right\|_2^2 \tag{112}$$

$$= L^2 \max_{j,k \in [h]} \left\| \theta_t^{(j)} - \theta_t^{(k)} \right\|_2^2 = L^2 \Delta_2\left(\vec{\theta}_t\right)^2. \tag{113}$$

Combining this with the previous lemmas completes the proof. □

**Lemma 11.** *Under assumptions (2, 3), we have*

$$\mathbb{E}_{\vec{\xi}_t | \vec{\theta}_t} \Delta_2\left(\vec{\theta}_{t+1}\right)^2 \le \left(\frac{1}{2} + L^2\eta^2\right) \Delta_2\left(\vec{\theta}_t\right)^2 + 4\sigma_t^2\eta^2 h. \tag{114}$$

*Proof.* We first bound the diameter of $\theta_{t+1/2}$, using Lemma 2 and the bound of Lemma 10. This yields

$$\mathbb{E}_{\vec{\xi}_t | \vec{\theta}_t} \Delta_2\left(\theta_{t+1/2}\right)^2 = \mathbb{E}_{\vec{\xi}_t | \vec{\theta}_t} \Delta_2\left(\vec{\theta}_t - \eta\vec{g}_t\right)^2 \le 2\Delta_2\left(\vec{\theta}_t\right)^2 + 2\Delta_2\left(\eta\vec{g}_t\right)^2 \tag{115}$$

$$= 2\Delta_2\left(\vec{\theta}_t\right)^2 + 2\eta^2\Delta_2\left(\vec{g}_t\right)^2 \le \left(2 + 4L^2\eta^2\right)\Delta_2\left(\vec{\theta}_t\right)^2 + 16\eta^2\sigma_t^2 h. \tag{116}$$

We now apply the asymptotic agreement guarantee of AVG, which yields

$$\mathop{\mathbb{E}}_{\vec{\xi}_t | \vec{\theta}_t} \Delta_2 \left( \vec{\theta}_{t+1} \right)^2 \leq \frac{1}{4} \mathop{\mathbb{E}}_{\vec{\xi}_t | \vec{\theta}_t} \Delta_2 \left( \vec{\theta}_{t+1/2} \right)^2 \tag{117}$$

$$\leq \frac{1}{4} \left( \left( 2 + 4L^2 \eta^2 \right) \Delta_2 \left( \vec{\theta}_t \right)^2 + 16 \eta^2 \sigma_t^2 h \right) \tag{118}$$

$$\leq \left( \frac{1}{2} + L^2 \eta^2 \right) \Delta_2 \left( \vec{\theta}_t \right)^2 + 4\eta^2 \sigma_t^2 h, \tag{119}$$

which is the lemma. $\square$

**Lemma 12.** *Under assumptions (2, 3), the diameter of the parameters is upper-bounded, i.e., there exists a constant $D$ such that for any $t \geq 1$ we have*

$$\mathop{\mathbb{E}}_{\xi_{1:t}} \Delta_2 \left( \vec{\theta}_t \right)^2 \leq D. \tag{120}$$

*Moreover, for any $\varepsilon > 0$, there exists an iteration $T^*(\varepsilon)$, such that for all $t \geq T^*(\varepsilon)$, we have*

$$\mathop{\mathbb{E}}_{\xi_{1:t}} \Delta_2 \left( \vec{\theta}_t \right)^2 \leq \varepsilon. \tag{121}$$

*Proof.* We know that $\eta \leq 1/2L$. Denoting $u_t \triangleq \mathbb{E} \Delta_2 \left( \vec{\theta}_t \right)^2$, by Lemma 11, we have

$$u_{t+1} \leq \frac{3}{4} u_t + 4\sigma_t^2 \eta^2 h. \tag{122}$$

By induction, we observe that, for all $t \geq 1$,

$$u_{t+1} \leq 4\eta^2 h \sum_{\tau=0}^{t-1} \left( \frac{3}{4} \right)^{\tau} \sigma_{t-\tau}^2. \tag{123}$$

Now recall that $\sigma_t$ is decreasing (see Section 2.2), thus, for all $t \geq 1$, we know $\sigma_t \leq \sigma_1$. Therefore,

$$u_{t+1} \leq 4\eta^2 h \sigma_1^2 \sum_{\tau=0}^{t-1} \left( \frac{3}{4} \right)^{\tau} \leq 4\eta^2 h \sigma_1^2 \sum_{\tau=0}^{\infty} \left( \frac{3}{4} \right)^{\tau} = 16 \eta^2 h \sigma_1^2 \triangleq D. \tag{124}$$

For the second part of the lemma, recall also from Section 2.2 that we regularly increase the batch size. Thus, there is an iteration $T_1(\varepsilon)$ such that for all $t \geq T_1(\varepsilon)$ we have $\sigma_t^2 \leq \frac{\varepsilon}{32\eta^2 h}$. By (123), we then have

$$u_{t+1} \leq 4\eta^2 h \sum_{\tau=0}^{t-T_1(\varepsilon)} \left( \frac{3}{4} \right)^{\tau} \sigma_{t-\tau}^2 + 4\eta^2 h \sum_{\tau=t-T_1(\varepsilon)+1}^{t-1} \left( \frac{3}{4} \right)^{\tau} \sigma_{t-\tau}^2 \tag{125}$$

$$\leq \frac{4\eta^2 h \varepsilon}{32\eta^2 h} \sum_{\tau=0}^{t-T_1(\varepsilon)} \left( \frac{3}{4} \right)^{\tau} + 4\eta^2 h \sigma_1^2 \sum_{\tau=t-T_1(\varepsilon)+1}^{t-1} \left( \frac{3}{4} \right)^{\tau} \tag{126}$$

$$\leq \frac{\varepsilon}{8} \sum_{\tau=0}^{\infty} \left( \frac{3}{4} \right)^{\tau} + 4\eta^2 h \sigma_1^2 \left( \frac{3}{4} \right)^{t-T_1(\varepsilon)+1} \sum_{\tau=0}^{\infty} \left( \frac{3}{4} \right)^{\tau} \tag{127}$$

$$\leq \frac{\varepsilon}{2} + 16 \eta^2 h \sigma_1^2 \left( \frac{3}{4} \right)^{t-T_1(\varepsilon)+1}. \tag{128}$$

Defining $T^*(\varepsilon) \triangleq T_1(\varepsilon) - 1 + \frac{\ln(32\eta^2 h \sigma_1^2 / \varepsilon)}{\ln(4/3)}$, we then have $u_{t+1} \leq \varepsilon$ for $t \geq T^*(\varepsilon)$, which is what we wanted.

$\square$

**Lemma 13.** *Under assumptions (2, 3), there exist constants $A$ and $B$, such that for all $t \geq 1$, we have*

$$\mathbb{E}_{\vec{\xi}_t | \vec{\theta}_t} \left\| \bar{G}_t - \nabla \mathcal{L}\left(\bar{\theta}_t\right) \right\|_2^2 \leq A\sigma_t^2 + B\Delta_2(\vec{\theta}_t)^2, \tag{129}$$

*where $\bar{G}_t$ is the average of the effective gradients of the nodes defined in (35).*

*Proof.* Note that

$$\bar{\theta}_{t+1} - \bar{\theta}_t = (\bar{\theta}_{t+1} - \bar{\theta}_{t+1/2}) + (\bar{\theta}_{t+1/2} - \bar{\theta}_t) \tag{130}$$

$$= (\bar{\theta}_{t+1} - \bar{\theta}_{t+1/2}) - \eta \bar{g}_t. \tag{131}$$

As a result, $\bar{G}_t = \bar{g}_t - \frac{1}{\eta}(\bar{\theta}_{t+1} - \bar{\theta}_{t+1/2})$. Moreover, we have

$$\bar{g}_t = \frac{1}{h} \sum_{j \in [h]} \nabla \mathcal{L}\left(\theta_t^{(j)}\right) + \frac{1}{h} \sum_{j \in [h]} \xi_t^{(j)} \tag{132}$$

$$= \nabla \mathcal{L}\left(\bar{\theta}_t\right) + \frac{1}{h} \sum_{j \in [h]} \left(\nabla \mathcal{L}\left(\theta_t^{(j)}\right) - \nabla \mathcal{L}\left(\bar{\theta}_t\right)\right) + \frac{1}{h} \sum_{j \in [h]} \xi_t^{(j)}. \tag{133}$$

This then yields:

$$\bar{G}_t - \nabla \mathcal{L}\left(\bar{\theta}_t\right) = \frac{1}{h} \sum_{j \in [h]} \left(\nabla \mathcal{L}\left(\theta_t^{(j)}\right) - \nabla \mathcal{L}\left(\bar{\theta}_t\right)\right) + \frac{1}{h} \sum_{j \in [h]} \xi_t^{(j)} + \frac{\bar{\theta}_{t+1/2} - \bar{\theta}_{t+1}}{\eta}. \tag{134}$$

Taking the $\ell_2$ norm on both sides and invoking Lemma 2 then implies that

$$\left\| \bar{G}_t - \nabla \mathcal{L}\left(\bar{\theta}_t\right) \right\|_2^2 \leq 3 \left\| \frac{1}{h} \sum_{j \in [h]} \left(\nabla \mathcal{L}\left(\theta_t^{(j)}\right) - \nabla \mathcal{L}\left(\bar{\theta}_t\right)\right) \right\|_2^2$$

$$+ 3 \left\| \frac{1}{h} \sum_{j \in [h]} \xi_t^{(j)} \right\|_2^2 + 3 \left\| \frac{\bar{\theta}_{t+1/2} - \bar{\theta}_{t+1}}{\eta} \right\|_2^2. \tag{135}$$

We now note that the expectation of each term can be bounded. Indeed,

$$\left\| \frac{1}{h} \sum_{j \in [h]} \left(\nabla \mathcal{L}\left(\theta_t^{(j)}\right) - \nabla \mathcal{L}\left(\bar{\theta}_t\right)\right) \right\|_2 \leq \frac{1}{h} \sum_{j \in [h]} \left\| \nabla \mathcal{L}\left(\theta_t^{(j)}\right) - \nabla \mathcal{L}\left(\bar{\theta}_t\right) \right\|_2 \tag{136}$$

$$\leq \frac{1}{h} \sum_{j \in [h]} L \left\| \theta_t^{(j)} - \bar{\theta}_t \right\|_2 \leq \frac{1}{h} \sum_{j \in [h]} L\Delta_2(\vec{\theta}_t) = L\Delta_2(\vec{\theta}_t). \tag{137}$$

Moreover,

$$\mathbb{E} \left\| \frac{1}{h} \sum_{j \in [h]} \xi_t^{(j)} \right\|_2^2 = \mathbb{E} \frac{1}{h^2} \sum_{j,k \in [h]} \xi_t^{(j)} \cdot \xi_t^{(k)} = \frac{1}{h^2} \sum_{j,k \in [h]} \mathbb{E} \, \xi_t^{(j)} \cdot \xi_t^{(k)} \tag{138}$$

$$= \frac{1}{h^2} \sum_{j \in [h]} \mathbb{E} \left\| \xi_t^{(j)} \right\|_2^2 \leq \frac{1}{h^2} \sum_{j \in [h]} \sigma_t^2 = \frac{\sigma_t^2}{h}, \tag{139}$$

using the fact that the noises are independent to move from one line to the other. For the last term, we use the $C$-averaging guarantee of AVG, yielding

$$\left\| \bar{\theta}_{t+1} - \bar{\theta}_{t+1/2} \right\|_2^2 \leq C^2 \Delta_2(\vec{\theta}_{t+1/2})^2. \tag{140}$$

To bound the right-hand side, note that we have $\vec{\theta}_{t+1/2} = \vec{\theta}_t - \eta \vec{g}_t = \vec{\theta}_t - \eta \overrightarrow{\nabla \mathcal{L}}(\vec{\theta}_t) - \eta \vec{\xi}_t$, where $\overrightarrow{\nabla \mathcal{L}}(\vec{\theta}_t) = \left( \nabla \mathcal{L}\left(\theta_t^{(1)}\right), \ldots, \nabla \mathcal{L}\left(\theta_t^{(h)}\right) \right)$. Lemma 2 then implies

$$\left\| \frac{\bar{\theta}_{t+1} - \bar{\theta}_{t+1/2}}{\eta} \right\|_2^2 \leq \frac{3C^2}{\eta^2} \Delta_2(\vec{\theta}_t)^2 + 3C^2 L^2 \Delta_2(\vec{\theta}_t)^2 + 3C^2 \Delta_2(\vec{\xi}_t)^2. \tag{141}$$

Combining it all, applying Lemma 9 and defining $B \triangleq \frac{9C^2}{\eta^2} + 3L^2 + 9C^2 L^2$, then yields

$$\mathbb{E}_{\vec{\xi}_t | \vec{\theta}_t} \left\| \bar{G}_t - \nabla \mathcal{L}\left(\bar{\theta}_t\right) \right\|_2^2 \leq \frac{3\sigma_t^2}{h} + \left( \frac{9C^2}{\eta^2} + 3L^2 + 9C^2 L^2 \right) \Delta_2(\vec{\theta}_t)^2 + 9C^2 \mathbb{E}_{\vec{\xi}_t | \vec{\theta}_t} \Delta_2(\vec{\xi}_t)^2 \tag{142}$$

$$\leq \left( \frac{3}{h} + 36C^2 h \right) \frac{\sigma_t^2}{h} + B \Delta_2(\vec{\theta}_t)^2. \tag{143}$$

Defining $A \triangleq \frac{3}{h} + 36C^2 h$ then yields the desired result. $\qquad \square$

We now proceed with the proof of our theorem.

*Proof of Theorem 3.* A direct consequence of Lipschitz continuity of the gradient of the loss function (Assumption 2) is that for all $\phi, \psi \in \mathbb{R}^d$, we have

$$\mathcal{L}(\psi) \leq \mathcal{L}(\phi) + (\psi - \phi) \cdot \nabla \mathcal{L}(\phi) + \frac{L}{2} \|\psi - \phi\|_2^2. \tag{144}$$

Therefore, by the definition of the effective gradient, we have

$$\mathcal{L}(\bar{\theta}_{t+1}) \leq \mathcal{L}(\bar{\theta}_t) - \eta \bar{G}_t \cdot \nabla \mathcal{L}\left(\bar{\theta}_t\right) + \frac{L\eta^2}{2} \left\| \bar{G}_t \right\|_2^2. \tag{145}$$

The fact that $\eta \leq 1/L$ then implies

$$\mathcal{L}(\bar{\theta}_{t+1}) \leq \mathcal{L}(\bar{\theta}_t) - \eta \bar{G}_t \cdot \nabla \mathcal{L}\left(\bar{\theta}_t\right) + \frac{\eta}{2} \left\| \bar{G}_t \right\|_2^2 \tag{146}$$

$$= \mathcal{L}(\bar{\theta}_t) - \frac{\eta}{2} \left\| \nabla \mathcal{L}\left(\bar{\theta}_t\right) \right\|_2^2 + \frac{\eta}{2} \left( \left\| \bar{G}_t \right\|_2^2 - 2\bar{G}_t \cdot \nabla \mathcal{L}\left(\bar{\theta}_t\right) + \left\| \nabla \mathcal{L}\left(\bar{\theta}_t\right) \right\|_2^2 \right) \tag{147}$$

$$= \mathcal{L}(\bar{\theta}_t) - \frac{\eta}{2} \left\| \nabla \mathcal{L}\left(\bar{\theta}_t\right) \right\|_2^2 + \frac{\eta}{2} \left\| \bar{G}_t - \nabla \mathcal{L}\left(\bar{\theta}_t\right) \right\|_2^2. \tag{148}$$

By rearranging the terms, we then have

$$\left\| \nabla \mathcal{L}\left(\bar{\theta}_t\right) \right\|_2^2 \leq \frac{2}{\eta} \left( \mathcal{L}(\bar{\theta}_t) - \mathcal{L}(\bar{\theta}_{t+1}) \right) + \left\| \bar{G}_t - \nabla \mathcal{L}\left(\bar{\theta}_t\right) \right\|_2^2. \tag{149}$$

Now taking the expectation over all of the stochastic noises and averaging over $t$, yields

$$\frac{1}{T} \sum_{t \in [T]} \mathbb{E}_{\vec{\xi}_{1:T}} \left\| \nabla \mathcal{L}\left(\bar{\theta}_t\right) \right\|_2^2 \leq \frac{2}{\eta T} \left( \mathcal{L}(\bar{\theta}_1) - \mathcal{L}(\bar{\theta}_{T+1}) \right) + \frac{1}{T} \sum_{t \in [T]} \mathbb{E}_{\vec{\xi}_{1:T}} \left\| \bar{G}_t - \nabla \mathcal{L}\left(\bar{\theta}_t\right) \right\|_2^2 \tag{150}$$

$$\leq \frac{2\mathcal{L}_{max}}{\eta T} + \frac{A}{T} \sum_{t \in [T]} \sigma_t^2 + \frac{B}{T} \sum_{t \in [T]} \mathbb{E}_{\vec{\xi}_{1:T}} \Delta_2(\vec{\theta}_t)^2, \tag{151}$$

where in the last inequality we used Lemma 13. The regular increase of the batch size (Section 2.2) then implies the existence of an iteration $T_1$, after which we have $\sigma_t^2 \leq \frac{\delta^2}{8A}$. Therefore, for $T \geq T_2 \triangleq \max \left\{ \frac{8\mathcal{L}_{max}}{\eta \delta^2}, T_1 \right\}$, we have

$$\frac{1}{T} \sum_{t \in [T]} \mathbb{E}_{\vec{\xi}_{1:T}} \left\| \nabla \mathcal{L}\left(\bar{\theta}_t\right) \right\|_2^2 \leq \frac{\delta^2}{4} + \frac{A}{T} \left( T_2 \sigma_1^2 + (T - T_2) \frac{\delta^2}{8A} \right) + \frac{B}{T} \sum_{t \in [T]} \mathbb{E}_{\vec{\xi}_{1:T}} \Delta_2(\vec{\theta}_t)^2 \tag{152}$$

$$\leq \frac{3\delta^2}{8} + \frac{T_2}{T} A \sigma_1^2 + \frac{B}{T} \sum_{t \in [T]} \mathbb{E}_{\vec{\xi}_{1:T}} \Delta_2(\vec{\theta}_t)^2. \tag{153}$$

For $T \geq T_3 \triangleq \frac{8T_2 A \sigma_1^2}{\delta^2}$, we then have

$$\frac{1}{T} \sum_{t \in [T]} \mathop{\mathbb{E}}_{\vec{\xi}_{1:T}} \left\| \nabla \mathcal{L} \left( \bar{\theta}_t \right) \right\|_2^2 \leq \frac{\delta^2}{2} + \frac{B}{T} \sum_{t \in [T]} \mathop{\mathbb{E}}_{\vec{\xi}_{1:T}} \Delta_2(\vec{\theta}_t)^2. \tag{154}$$

We now invoke Lemma 12 with $\varepsilon = \frac{1}{2} \min \left\{ \delta^2, \delta^2/2B \right\}$. For $T \geq T^*(\varepsilon)$ we then have

$$\frac{1}{T} \sum_{t \in [T]} \mathop{\mathbb{E}}_{\vec{\xi}_{1:T}} \Delta_2(\vec{\theta}_t)^2 = \frac{1}{T} \sum_{t=1}^{T^*(\varepsilon)} \mathop{\mathbb{E}}_{\vec{\xi}_{1:T}} \Delta_2(\vec{\theta}_t)^2 + \frac{1}{T} \sum_{t=T^*(\varepsilon)+1}^{T} \mathop{\mathbb{E}}_{\vec{\xi}_{1:T}} \Delta_2(\vec{\theta}_t)^2 \tag{155}$$

$$\leq \frac{T^*(\varepsilon)}{T} D + \frac{T - T^*(\varepsilon)}{T} \varepsilon \leq \frac{T^*(\varepsilon)}{T} D + \varepsilon. \tag{156}$$

Thus, for $T \geq T_4 \triangleq \frac{D T^*(\varepsilon)}{\varepsilon}$ we have

$$\mathbb{E} \, \Delta_2(\vec{\theta}_*)^2 = \frac{1}{T} \sum_{t \in [T]} \mathop{\mathbb{E}}_{\vec{\xi}_{1:T}} \Delta_2(\vec{\theta}_t)^2 \leq 2\varepsilon = \min \left\{ \delta^2, \delta^2/2B \right\}. \tag{157}$$

Combining this with (154), for $T = T_{\text{HOM-LEARN}}(\delta) \triangleq \max \{T_3, T_4\}$ we then have

$$\mathbb{E} \, \Delta_2(\vec{\theta}_*)^2 \leq \delta^2 \quad \text{and} \quad \mathbb{E} \left\| \nabla \mathcal{L} \left( \bar{\theta}_* \right) \right\|_2^2 \leq \delta^2, \tag{158}$$

which is the desired result.

$\square$

## C  MDA Algorithm

### C.1  Correctness proof of MDA

We first note a few important properties of MDA.

**Lemma 14.** *The $\ell_2$ diameter of the* MDA *subfamily is upper-bounded by that of the honest vectors. In other words, for any Byzantine attack $\overrightarrow{\text{BYZ}}$, denoting $\vec{z} \triangleq \overrightarrow{\text{BYZ}}(\vec{x})$, we have*

$$\Delta_2 \left( \vec{z}^{(\text{MDA})} \right) \leq \Delta_2 \left( \vec{x} \right) \tag{159}$$

*Proof.* Since $\overrightarrow{\text{BYZ}}$ selects $q$ vectors, out of which at most $f$ are Byzantine vectors, we know that there exists a subset $H \subset [q]$ of cardinal at least $q - f$ that only contains honest vectors. But then, we have

$$\Delta_2 \left( \vec{z}^{(\text{MDA})} \right) = \min_{\substack{S \subset [q] \\ |S| = q-f}} \Delta_2 \left( \vec{z}^{(S)} \right) \leq \Delta_2 \left( \vec{z}^{(H)} \right) \leq \Delta_2(\vec{x}), \tag{160}$$

which is the lemma. $\square$

**Lemma 15.** *Under Assumption 5,* MDA *guarantees Byzantine asymptotic agreement. In other words, for any input $N \in \mathbb{N}$ and any family $\vec{x} \in \mathbb{R}^{d \cdot h}$, denoting $\vec{x}_N \triangleq \overrightarrow{\text{MDA}}_N \circ \overrightarrow{\text{BYZ}}_N(\vec{x})$,*

$$\Delta_2(\vec{x}_N) \leq \frac{\Delta_2(\vec{x})}{2^N}. \tag{161}$$

*Proof.* Denote $\vec{z}^{(1)} \triangleq \overrightarrow{\text{BYZ}}^{(1)}(\vec{x})$ and $\vec{z}^{(2)} \triangleq \overrightarrow{\text{BYZ}}^{(2)}(\vec{x})$ the results of the two Byzantine attacks, $S_1 = S_{\text{MDA}}(\vec{z}^{(1)})$ and $S_2 = S_{\text{MDA}}(\vec{z}^{(2)})$ the subsets selected by MDA in the two cases.

Moreover, we write $S_1 = H_1 \cup F_1$ and $S_2 = H_2 \cup F_2$, where $H_1$ and $H_2$ are subsets of honest vectors within $S_1$ and $S_2$. Without loss of generality, we assume both $H_1$ and $H_2$ to be of cardinal $q - 2f$. As a result, we know that there exist injective functions $\sigma_1 : H_1 \to [h]$ and $\sigma_2 : H_2 \to [h]$ such that $z^{(1,j)} = x^{(\sigma_1(j))}$ and $z^{(2,k)} = x^{(\sigma_2(k))}$, for all $j \in H_1$ and $k \in H_2$.

Finally, we denote $y^{(1)} \triangleq \text{MDA}(\vec{z}^{(1)})$ and $y^{(2)} \triangleq \text{MDA}(\vec{z}^{(2)})$. We then have

$$(q-f)\left\|y^{(1)} - y^{(2)}\right\|_2 = \left\|\sum_{j\in S_1} z^{(1,j)} - \sum_{k\in S_2} z^{(2,k)}\right\|_2 \tag{162}$$

$$= \left\|\sum_{j\in F_1} z^{(1,j)} - \sum_{k\in F_2} z^{(2,k)} + \sum_{j\in\sigma_1(H_1)} x^{(j)} - \sum_{k\in\sigma_2(H_2)} x^{(k)}\right\|_2 \tag{163}$$

$$\leq \left\|\sum_{j\in F_1} z^{(1,j)} - \sum_{k\in F_2} z^{(2,k)}\right\|_2 + \left\|\sum_{j\in\sigma_1(H_1)-\sigma_2(H_2)} x^{(j)} - \sum_{k\in\sigma_2(H_2)-\sigma_1(H_1)} x^{(k)}\right\|_2. \tag{164}$$

Note that $|F_1| = |S_1 - H_1| = f = |S_2 - H_2| = |F_2|$. Moreover,

$$|\sigma_1(H_1) - \sigma_2(H_2)| = |\sigma_1(H_1) \cup \sigma_2(H_2) - \sigma_2(H_2)| \tag{165}$$

$$= |\sigma_1(H_1) \cup \sigma_2(H_2)| - |\sigma_2(H_2)| \leq |[h]| - |H_2| = 2f + h - q, \tag{166}$$

and similarly for $\sigma_2(H_2)-\sigma_1(H_1)$. Note that $|\sigma_1(H_1)|, |\sigma_2(H_2)| \geq q-2f$. Therefore, Assumption 5 implies that

$$|\sigma_1(H_1)| + |\sigma_2(H_2)| \geq 2\left(\frac{1+\varepsilon}{2}h + \frac{5+3\varepsilon}{2}f - 2f\right) > h, \tag{167}$$

which yields $\sigma_1(H_1) \cap \sigma_2(H_2) \neq \varnothing$. Now let $\gamma$ be an element of the intersection of $\sigma_1(H_1)$ and $\sigma_2(H_2)$, and consider any bijections $\tau_F : F_1 \to F_2$ and $\tau_H : \sigma_1(H_1)-\sigma_2(H_2) \to \sigma_2(H_2)-\sigma_1(H_1)$. Using triangle inequality and Lemma 14, for any $j \in F_1$, we then have

$$\left\|z^{(1,j)} - z^{(2,\tau_F(j))}\right\|_2 \leq \left\|z^{(1,j)} - x^{(\gamma)}\right\|_2 + \left\|x^{(\gamma)} - z^{(2,\tau_F(j))}\right\|_2 \leq 2\Delta_2(\vec{x}). \tag{168}$$

Combining it all, we obtain

$$(q-f)\left\|y^{(1)} - y^{(2)}\right\|_2 \leq \sum_{j\in F_1}\left\|z^{(1,j)} - z^{(2,\tau_F(j))}\right\|_2 + \sum_{j\in\sigma_1(H_1)-\sigma_2(H_2)}\left\|x^{(j)} - x^{(\tau_H(j))}\right\|_2 \tag{169}$$

$$\leq 2f\Delta_2(\vec{x}) + (2f + h - q)\Delta_2(\vec{x}), \tag{170}$$

which implies

$$\left\|y^{(1)} - y^{(2)}\right\|_2 \leq \frac{4f + h - q}{q - f}\Delta_2(\vec{x}). \tag{171}$$

We then apply Assumption 5, which implies that

$$\frac{4f + h - q}{q - f} \leq \frac{4f + h - \frac{1+\varepsilon}{2}h - \frac{5+3\varepsilon}{2}f}{\frac{1+\varepsilon}{2}h + \frac{5+3\varepsilon}{2}f - f} = \frac{(1-\varepsilon)h + 3(1-\varepsilon)f}{(1+\varepsilon)h + 3(1+\varepsilon)f} \tag{172}$$

$$= \frac{1-\varepsilon}{1+\varepsilon} = \frac{1+\varepsilon - 2\varepsilon}{1+\varepsilon} = 1 - \frac{2\varepsilon}{1+\varepsilon} = 1 - \tilde{\varepsilon}. \tag{173}$$

This shows that $\Delta_2(\vec{y}) \leq (1 - \tilde{\varepsilon})\Delta_2(\vec{x})$. In other words, one iteration of MDA is guaranteed to multiply the $\ell_2$ diameter of honest nodes by at most $(1 - \tilde{\varepsilon})$. It follows that $T_{\text{MDA}}(N) = \lceil N \ln 2/\tilde{\varepsilon}\rceil$ iterations will multiply this diameter by at most $(1 - \tilde{\varepsilon})^{T_{\text{MDA}}(N)} \leq \exp\left(\frac{\ln(1-\tilde{\varepsilon})\ln 2}{\tilde{\varepsilon}}\right)^N \leq 2^{-N}$, using the inequality $\ln(1 - \tilde{\varepsilon}) \leq -\tilde{\varepsilon}$. $\qquad\square$

**Remark 4.** *Note that we can set $\varepsilon = \frac{n-6f}{n+2f}$. Thus, in the regime $f \ll n$, we have $\varepsilon \to 1$, which implies $\tilde{\varepsilon} \to 1$. Thus, for any fixed value of $N$, given that our proof showed that $\Delta_2(\vec{y}) \leq (1 - \tilde{\varepsilon})\Delta_2(\vec{x})$, when $f$ is sufficiently small compared to $n$, MDA actually achieves asymptotic agreement in only one communication round.*

**Lemma 16.** *One iteration of MDA returns a vector close to the average of the honest vectors. Denoting $\vec{y} \triangleq \overrightarrow{\text{MDA}} \circ \overrightarrow{\text{BYZ}}(\vec{x})$, we have*

$$\|\bar{y} - \bar{x}\|_2 \leq \frac{(2f + h - q)q + (q - 2f)f}{h(q - f)}\Delta_2(\vec{x}). \tag{174}$$

*In the synchronous case where $q = n = f + h$, the right-hand side becomes $\frac{2f}{h}\Delta_2(\vec{x})$.*

*Proof.* Let us write $S_{\mathrm{MDA}}(\vec{z}) = H \cup F$, where $H$ are honest vectors and $F$ are Byzantine vectors. We know that $|H| \geq q - 2f$ and $|H| + |F| = q - f$. In fact, without loss of generality, we can assume $|H| = q - 2f$ (since this is equivalent to labeling honest vectors not in $H$ as Byzantine vectors).

Let us also denote $\sigma : H \to [h]$ the injective function that maps honest vectors to the index of their node, and $\bar{H} = [h] - \sigma(H)$ the unqueried nodes. We have

$$\|y - \bar{x}\|_2 = \left\| \frac{|H|\,\bar{z}^{(H)} + |F|\,\bar{z}^{(F)}}{|H| + |F|} - \frac{|\sigma(H)|\,\bar{x}^{(\sigma(H))} + |\bar{H}|\,\bar{x}^{(\bar{H})}}{|\sigma(H)| + |\bar{H}|} \right\|_2 \tag{175}$$

$$= \left\| \frac{|H|\,\bar{z}^{(H)} + |F|\,\bar{z}^{(F)}}{|H| + |F|} - \frac{|H|\,\bar{z}^{(H)} + |\bar{H}|\,\bar{x}^{(\bar{H})}}{|H| + |\bar{H}|} \right\|_2 \tag{176}$$

$$= \frac{\left\| |H|\left(|\bar{H}| - |F|\right)\bar{z}^{(H)} + |F|\left(|H| + |\bar{H}|\right)\bar{z}^{(F)} - |\bar{H}|\left(|H| + |F|\right)\bar{x}^{(\bar{H})} \right\|_2}{\left(|H| + |F|\right)\left(|H| + |\bar{H}|\right)} \tag{177}$$

$$= \frac{\left\| |H||\bar{H}|\left(\bar{z}^{(H)} - \bar{x}^{(\bar{H})}\right) + |F||H|\left(\bar{z}^{(H)} - \bar{z}^{(F)}\right) + |\bar{H}||F|\left(\bar{z}^{(F)} - \bar{x}^{(\bar{H})}\right) \right\|_2}{\left(|H| + |F|\right)\left(|H| + |\bar{H}|\right)} \tag{178}$$

$$\leq \frac{|H||\bar{H}|\left\|\bar{z}^{(H)} - \bar{x}^{(\bar{H})}\right\|_2 + |F||H|\left\|\bar{z}^{(H)} - \bar{z}^{(F)}\right\|_2 + |\bar{H}||F|\left\|\bar{z}^{(F)} - \bar{x}^{(\bar{H})}\right\|_2}{\left(|H| + |F|\right)\left(|H| + |\bar{H}|\right)}. \tag{179}$$

Now note that

$$\left\|\bar{z}^{(H)} - \bar{x}^{(\bar{H})}\right\|_2 = \left\|\bar{x}^{(\sigma(H))} - \bar{x}^{(\bar{H})}\right\|_2 \tag{180}$$

$$= \frac{1}{|\sigma(H)||\bar{H}|}\left\| |\bar{H}| \sum_{j \in \sigma(H)} x^{(j)} - |H| \sum_{k \in \bar{H}} x^{(k)} \right\|_2 \tag{181}$$

$$= \frac{1}{|\sigma(H)||\bar{H}|}\left\| \sum_{k \in \bar{H}} \sum_{j \in \sigma(H)} x^{(j)} - \sum_{j \in \sigma(H)} \sum_{k \in \bar{H}} x^{(k)} \right\|_2 \tag{182}$$

$$= \frac{1}{|\sigma(H)||\bar{H}|}\left\| \sum_{j \in \sigma(H)} \sum_{k \in \bar{H}} \left(x^{(j)} - x^{(k)}\right) \right\|_2 \tag{183}$$

$$\leq \frac{1}{|\sigma(H)||\bar{H}|} \sum_{j \in \sigma(H)} \sum_{k \in \bar{H}} \left\|x^{(j)} - x^{(k)}\right\|_2 \tag{184}$$

$$\leq \frac{1}{|\sigma(H)||\bar{H}|} \sum_{j \in \sigma(H)} \sum_{k \in \bar{H}} \Delta_2(\vec{x}) = \Delta_2(\vec{x}). \tag{185}$$

Similarly, we show that $\left\|\bar{z}^{(H)} - \bar{z}^{(F)}\right\|_2 \leq \Delta_2(\bar{z}^{(\mathrm{MDA})}) \leq \Delta_2(\vec{x})$. Finally, we use the triangle inequality to show that

$$\left\|\bar{z}^{(F)} - \bar{x}^{(\bar{H})}\right\|_2 \leq \left\|\bar{z}^{(F)} - \bar{z}^{(H)}\right\|_2 + \left\|\bar{z}^{(H)} - \bar{x}^{(\bar{H})}\right\|_2 \leq 2\Delta_2(\vec{x}). \tag{186}$$

Therefore, we now have

$$\|y - \bar{x}\|_2 \leq \frac{|H||\bar{H}| + |F||H| + 2|\bar{H}||F|}{\left(|H| + |F|\right)\left(|H| + |\bar{H}|\right)} \Delta_2(\vec{x}) \tag{187}$$

$$= \frac{(q - 2f)(2f + h - q) + f(q - 2f) + 2(2f + h - q)f}{h(q - f)} \Delta_2(\vec{x}) \tag{188}$$

$$= \frac{(2f + h - q)q + (q - 2f)f}{h(q - f)} \Delta_2(\vec{x}), \tag{189}$$

which is the lemma. $\qquad\square$

Now we can prove our theorem.

*Proof of Theorem 4.* Lemma 15 already proved asymptotic agreement. Moreover, using Lemma 16, and denoting $\alpha \triangleq \frac{(2f+h-q)q+(q-2f)f}{h(q-f)}$ and $\vec{x}_t$ the vector family obtained after $t$ iterations of MDA, we also know that

$$\left\| \bar{x}_{t+1} - \bar{x}_t \right\|_2 \leq \alpha \Delta_2(\vec{x}_t) \leq \alpha(1-\tilde{\varepsilon})^t \Delta_2(\vec{x}_0). \tag{190}$$

Using triangle inequality then yields, for any number $T_{\mathrm{MDA}}(N)$ of iterations of MDA,

$$\left\| \bar{x}_{T_{\mathrm{MDA}}(N)} - \bar{x}_0 \right\|_2 \leq \sum_{t=0}^{T_{\mathrm{MDA}}(N)-1} \left\| \bar{x}_{t+1} - \bar{x}_t \right\|_2 \leq \sum_{t=0}^{T_{\mathrm{MDA}}(N)-1} \alpha(1-\tilde{\varepsilon})^t \Delta_2(\vec{x}_0) \tag{191}$$

$$\leq \alpha \Delta_2(\vec{x}_0) \sum_{t=0}^{\infty} (1-\tilde{\varepsilon})^t = \frac{\alpha \Delta_2(\vec{x}_0)}{\tilde{\varepsilon}}, \tag{192}$$

which is the guarantee of the theorem. $\qquad\square$

### C.2 Lower bound on the averaging constant

We prove here a lower bound on the averaging constant that any algorithm can achieve. Our proof requires the construction of hard instances. We use the following notation.

**Definition 4** ($\star$ notation). *We denote by $x \star h \triangleq \big( \underbrace{x, \ldots, x}_{h \text{ times}} \big)$ the repetition of a value $x$ $h$ times.*

**Lemma 17** (Quasi-unanimity). *For any averaging agreement algorithm* AVG, *whenever a node $j$ only hears from $q$ nodes, assuming $q - f$ of these nodes act like honest nodes with the same initial value $x$, then* AVG *must make node $j$ output $x$.*

*Proof.* For any agreeing initial family $\vec{x} = x \star h$, we have $\Delta_2(\vec{x}) = 0$ and $\bar{x} = x$. Then averaging agreement implies that, for any $N \in \mathbb{N}$, we output $\vec{x}_N$ such that

$$\Delta_2(\vec{x}_N) \leq \frac{\Delta_2(\vec{x})}{2^N} = 0 \quad \text{and} \quad \left\| \bar{x}_N - x \right\|_2 \leq C\Delta_2(\vec{x}) = 0. \tag{193}$$

In other words, we must have $\vec{x}_N = \vec{x}$.

But then, if node $j$ only hears from $q$ nodes, and if it receives $q - f$ nodes agreeing on a value $x$, then it cannot exclude the possibility that the remaining $f$ nodes come from Byzantine nodes. As a result, node $j$ cannot exclude that the initial family was $\vec{x}$. To satisfy averaging agreement, node $j$ must then output $x$. $\qquad\square$

*Proof of Theorem 5.* Consider the vector family defined by

$$\vec{x} \triangleq \left( 0 \star (q - 2f), 1 \star (h + 2f - q) \right). \tag{194}$$

For any algorithm AVG used by honest nodes, Byzantine nodes can slow down all messages from nodes in $[q - f + 1, h]$ to nodes in $[q - f]$. Thus, the first $q - f$ honest nodes would be making decisions without receiving any input from nodes in $[q - f + 1, h]$. Assume now that the Byzantine nodes all act exactly like the first $q - 2f$ nodes. Then, all first $q - f$ nodes would see $q - f$ nodes acting like honest nodes with initial vector $0$, and $f$ nodes acting like honest nodes with initial vector $1$. By quasi-unanimity (Lemma 17), the $q - 2f$ first nodes must output $0$.

But now, by asymptotic agreement, this implies that any other honest node must output a vector at distance at most $\Delta_2(\vec{x})/2^N = 1/2^N$ of $0$. As a result, as $N \to \infty$, denoting $\vec{x}_N$ the output of AVG for input $N \in \mathbb{N}$, we must have $\bar{x}_N \to 0$.

Since $\Delta_2(\vec{x}) = 1$ and $\bar{x} = (h + 2f - q)/h$, we then have

$$\lim_{N \to 0} |\bar{x}_N - \bar{x}| = |0 - \bar{x}| = \frac{h + 2f - q}{h} \geq \frac{h + 2f - q}{h} \Delta_2(\vec{x}). \tag{195}$$

This shows that AVG cannot achieve better than $\frac{h+2f-q}{h}$-averaging agreement which is equal to $\frac{2f}{h}$, for $q = h$.

Now we show MDA indeed achieves this bound up to a multiplicative constant. From Assumption 5, we can set $\varepsilon = \frac{n-6f}{n+2f} = \frac{h-5f}{h+3f}$. In the regime $q = h$ and $f \ll h$, we thus have $\varepsilon \to 1$, which then implies $\tilde{\varepsilon} \to 1$. Now notice that

$$\frac{(2f+h-q)q + (q-2f)f}{h(q-f)\tilde{\varepsilon}} = \frac{2f/h}{(1-\frac{f}{q})\tilde{\varepsilon}} + \frac{f}{h}\frac{1-\frac{2f}{q}}{(1-\frac{f}{q})\tilde{\varepsilon}} = \frac{3f}{h} + o(1). \tag{196}$$

Yet, for $q = h$, we showed that the lower bound on the averaging constant is $2f/h$. MDA is thus asymptotically optimal, up to the multiplicative constant $3/2$. $\qquad\square$

### C.3   Note on Byzantine tolerance

Our MDA algorithm tolerates a small fraction of Byzantine nodes, namely $n > 6f$.

**Proposition 1.** *Assume $n \leq 6f$. For any parameter $N$, no matter how the number $T_{\mathrm{MDA}}(N)$ of iterations is chosen, Byzantines can make* MDA *fail to achieve asymptotic agreement. As a result,* MDA *cannot guarantee averaging agreement for $n \leq 6f$.*

*Proof.* Define $\delta \triangleq \min\left\{1, 4 - 4 \cdot 2^{-(N-1)/T_{\mathrm{MDA}}(N)}\right\}$, and consider the honest vector family

$$\vec{x} \triangleq (-1 \star 2f, 0 \star f, 1 \star 2f). \tag{197}$$

For the first $2f$ nodes, Byzantine nodes can block $f$ of the messages of the last $2f$ nodes, and add $f$ values equal to $-2 + \delta$. The first $2f$ honest nodes then observe

$$\vec{z}_1 \triangleq ((-2+\delta) \star f, -1 \star 2f, 0 \star f, 1 \star f). \tag{198}$$

MDA would then remove the largest $f$ inputs, as it then achieves a diameter equal to $2 - \delta < 2$. The first $2f$ nodes would then output

$$\frac{(-2+\delta)f - 2f}{4f} = -\frac{4f - \delta f}{4f} = -1 + \frac{\delta}{4}. \tag{199}$$

For the middle $f$ nodes, Byzantine nodes can simply allow perfect communication and not intervene. By symmetry, the middle $f$ nodes would output $0$ (note that to be rigorous, we could define MDA as taking the average of the outputs over all subsets of inputs of minimal diameter).

For the last $2f$ nodes, Byzantine nodes can block $f$ messages of the first $2f$ nodes, and add $f$ values equal to $2 - \delta$. The situation is then symmetric to the case for the first $2f$ nodes, and make the last $2f$ nodes output $1 - \frac{\delta}{4}$.

As a result, one iteration of MDA multiplies the diameter of the honest nodes by $1 - \frac{\delta}{4} \geq 2^{-(N-1)/T_{\mathrm{MDA}}(N)}$. Byzantine nodes can use the same strategy in all other iterations. Thus, denoting $\vec{y} \triangleq \overrightarrow{\mathrm{MDA}} \circ \overrightarrow{\mathrm{BYZ}}(\vec{x})$, we have

$$\Delta_2(\vec{y}) = \left(1 - \frac{\delta}{4}\right)^T \Delta_2(\vec{x}) \geq \left(2^{-(N-1)/T_{\mathrm{MDA}}(N)}\right)^{T_{\mathrm{MDA}}(N)} \Delta_2(\vec{x}) = 2\frac{\Delta_2(\vec{x})}{2^N} > \frac{\Delta_2(\vec{x})}{2^N}, \tag{200}$$

which shows that the final vectors obtained by MDA violate asymptotic agreement. $\qquad\square$

## D   RB-TM Algorithm

In this section, we prove that RB-TM solves averaging agreement under Assumption 6.

### D.1   Diameters

Before doing so, we introduce the notion of $\ell_r$-diameters and we prove a few useful lemmas.

**Definition 5.** *For any $r \in [1, \infty]$, we define the diameter along coordinate $i$ by*

$$\Delta^{cw}(\vec{x})[i] = \max_{j,k \in [h]} \left| x^{(j)}[i] - x^{(k)}[i] \right|, \tag{201}$$

*and the coordinate-wise $\ell_r$-diameters by $\Delta_r^{cw}(\vec{x}) = \|\Delta^{cw}(\vec{x})\|_r$.*

Interestingly, we have the following bounds between diameters.

**Lemma 18.** *The $\ell_r$-diameters are upper-bounded by coordinate-wise $\ell_r$-diameters, i.e.,*

$$\forall r, \ \Delta_r \leq \Delta_r^{cw} \leq \min\left\{d^{1/r}, 2h^{1/r}\right\}\Delta_r. \tag{202}$$

*Note that in ML applications, we usually expect $d \gg h$, in which case the more relevant right-hand side inequality is $\Delta_r^{cw} \leq 2h^{1/r}\Delta_r$.*

*Proof.* Consider $j^*, k^* \in [h]$ such that $\Delta_r(\vec{x}) = \left\|x^{(j^*)} - x^{(k^*)}\right\|_r$. But then, we note that on each coordinate $i \in [d]$,

$$\left|x^{(j^*)}[i] - x^{(k^*)}[i]\right| \leq \max_{j,k \in [h]} \left|x^{(j)}[i] - x^{(k)}[i]\right| = \Delta^{cw}(x)[i]. \tag{203}$$

As a result, $\Delta_r(\vec{x}) = \left\|x^{(j^*)} - x^{(k^*)}\right\|_r \leq \left\|\Delta^{cw}(\vec{x})\right\|_r = \Delta_r^{cw}(\vec{x})$. For the right-hand side, first note that a coordinate-wise diameter is smaller than the $\ell_r$ diameter, which yields

$$\Delta_r^{cw}(\vec{x})^r = \sum_{i \in [d]} \max_{j,k \in [h]} \left|x^{(j)}[i] - x^{(k)}[i]\right|^r \leq \sum_{i \in [d]} \max_{j,k \in [h]} \left\|x^{(j)} - x^{(k)}\right\|_r^r \tag{204}$$

$$= \sum_{i \in [d]} \Delta_r(\vec{x})^r = d\Delta_r(\vec{x})^r. \tag{205}$$

Taking the $r$-th root shows that $\Delta_r^{cw} \leq d^{1/r}\Delta_r$. What is left to prove is that $\Delta_r^{cw} \leq 2h^{1/r}\Delta_r$. To prove this, note that for any $i \in [d]$, we have

$$\max_{j,k \in [h]} \left|x^{(j)}[i] - x^{(k)}[i]\right| \leq \max_{j \in [h]} \left(2\left|x^{(j)}[i] - x^{(1)}[i]\right|\right). \tag{206}$$

Indeed, assuming the former maximum is reached for $j^*$ and $k^*$, the latter maximum will be reached for $j^*$ or $k^*$, depending on whether $x^{(1)}[i]$ is closer to $x^{(j^*)}[i]$ or $x^{(k^*)}[i]$. In either case, the above inequality holds. As a result,

$$\Delta_r^{cw}(\vec{x})^r = \sum_{i \in [d]} \max_{j,k \in [h]} \left|x^{(j)}[i] - x^{(k)}[i]\right|^r \leq \sum_{i \in [d]} \max_{j \in [h]} \left(2\left|x^{(j)}[i] - x^{(1)}[i]\right|\right)^r \tag{207}$$

$$= 2^r \sum_{i \in [d]} \max_{j \in [h]} \left|x^{(j)}[i] - x^{(1)}[i]\right|^r \leq 2^r \sum_{i \in [d]} \sum_{j \in [h]} \left|x^{(j)}[i] - x^{(1)}[i]\right|^r \tag{208}$$

$$= 2^r \sum_{j \in [h]} \sum_{i \in [d]} \left|x^{(j)}[i] - x^{(1)}[i]\right|^r = 2^r \sum_{j \in [h]} \left\|x^{(j)} - x^{(1)}\right\|_r^r \tag{209}$$

$$\leq 2^r \sum_{j \in [h]} \Delta_r(\vec{x})^r = 2^r h \Delta_r(\vec{x})^r. \tag{210}$$

Taking the $r$-th root yields $\Delta_r^{cw} \leq 2h^{1/r}\Delta_r$, which concludes the proof. $\qquad\square$

As an immediate corollary, asymptotic agreement is equivalent to showing that *any* of the diameters we introduce in this section goes to zero.

Interestingly, our diameters satisfy the triangle inequality, as shown by the following lemma.

**Lemma 19.** *The diameters and coordinate-wise diameters satisfy the triangle inequality. Namely, for any two families of vectors $\vec{x}$ and $\vec{y}$, we have the following inequality*

$$\Delta^{cw}(\vec{x} + \vec{y}) \leq \Delta^{cw}(\vec{x}) + \Delta^{cw}(\vec{y}). \tag{211}$$

*As an immediate corollary, by triangle inequality of norms, for any $r \in [1, \infty]$, we also have $\Delta_r^{cw}(\vec{x} + \vec{y}) \leq \Delta_r^{cw}(\vec{x}) + \Delta_r^{cw}(\vec{y})$. We also have $\Delta_r(\vec{x} + \vec{y}) \leq \Delta_r(\vec{x}) + \Delta_r(\vec{y})$.*

*Proof.* For any coordinate $i \in [d]$, the following holds:

$$\Delta^{cw}(\vec{x} + \vec{y})[i] = \max_{j,k \in [h]} \left| x^{(j)}[i] + y^{(j)}[i] - x^{(k)}[i] - y^{(k)}[i] \right| \tag{212}$$

$$\leq \max_{j,k \in [h]} \left\{ \left| x^{(j)}[i] - x^{(k)}[i] \right| + \left| y^{(j)}[i] - y^{(k)}[i] \right| \right\} \tag{213}$$

$$\leq \max_{j,k \in [h]} \left| x^{(j)}[i] - x^{(k)}[i] \right| + \max_{j',k' \in [h]} \left| y^{(j')}[i] - y^{(k')}[i] \right| \tag{214}$$

$$= \Delta^{cw}(\vec{x})[i] + \Delta^{cw}(\vec{y})[i], \tag{215}$$

which concludes the proof for coordinate-wise diameters. The proof for $\ell_r$ diameters is similar. $\square$

## D.2 Correctness proof of RB-TM and Lower bound on Byzantine tolerance

We now move on to the proof of correctness of RB-TM for $n \geq 3f + 1$. First, denoting $S^{(j)}[i] = S(\vec{z}^{(j)}[i])$, note that we have the following lemma.

**Lemma 20.** *For any two honest nodes $j$ and $k$ and any coordinate $i$, we have*

$$\left| S^{(j)}[i] - S^{(k)}[i] \right| \leq f. \tag{216}$$

*Proof.* Note that node $j$ receives messages from at most $n$ nodes, and in the trimming step, $2f$ nodes are discarded, which yields $\left| S^{(j)}[i] \right| \leq n - 2f$. Moreover, among the nodes in $Q^{(j)} \cap Q^{(k)}$ at most, $f$ nodes with the smallest and $f$ nodes with the largest $i$-th coordinates will be trimmed. Now recall that $\left| Q^{(j)} \cap Q^{(k)} \right| \geq q$, thus, $\left| S^{(j)}[i] \cap S^{(k)}[i] \right| \geq q - 2f = n - 3f$. We then obtain

$$\left| S^{(j)}[i] - S^{(k)}[i] \right| = \left| S^{(j)}[i] \right| - \left| S^{(j)}[i] \cap S^{(k)}[i] \right| \leq f, \tag{217}$$

which is the lemma. $\square$

We denote $x^{(min)}[i] = \min_{j \in [h]} x^{(j)}[i]$ and $x^{(max)}[i] = \max_{j \in [h]} x^{(j)}[i]$ the minimal and maximal $i$-th coordinate among the parameters of the honest nodes.

**Lemma 21.** *All the values that are not discarded in the trimming step are within the range of the values proposed by the honest nodes, i.e.,*

$$\forall i \in [d], \ \forall j \in [h], \ \forall k \in S^{(j)}[i], \ x^{(min)}[i] \leq w^{(k)}[i] \leq x^{(max)}[i] \tag{218}$$

*Proof.* Note that there exist at most $f$ Byzantine nodes that might broadcast vectors with the $i$-th coordinate larger than $x^{(max)}[i]$ or smaller than $x^{(min)}[i]$, and all of these nodes will be removed by trimming. $\square$

**Lemma 22** (Contraction by TM). *Under Assumption 6, TM guarantees the contraction of the coordinate-wise diameters, that is,*

$$\forall \vec{x}, \ \forall \overrightarrow{BYZ}, \Delta^{cw} \left( \overrightarrow{TM} \circ \overrightarrow{BYZ} (\vec{x}) \right) \leq (1 - \tilde{\varepsilon}) \Delta^{cw} (\vec{x}). \tag{219}$$

*As an immediate corollary, this inequality holds by taking the $\ell_r$-norm on both sides, which means that the coordinate-wise $\ell_r$-diameter is also contracted by the same factor.*

*Proof.* Let us bound the distance between $y^{(j)}[i]$ and $y^{(k)}[i]$, the $i$-th coordinate of the outputs of two arbitrary nodes $j$ and $k$ after a Byzantine attack. Denote

$$m = \frac{1}{\left| S^{(j)}[i] \cap S^{(k)}[i] \right|} \sum_{l \in S^{(j)}[i] \cap S^{(k)}[i]} w^{(l)}[i], \tag{220}$$

the average of the $i$-th coordinates of the nodes in $S^{(j)}[i] \cap S^{(k)}[i]$. Without loss of generality, assume $y^{(j)}[i] \geq y^{(k)}[i]$. We then obtain

$$\left|y^{(j)}[i] - y^{(k)}[i]\right| = \frac{1}{\left|S^{(j)}[i]\right|} \sum_{l \in S^{(j)}[i]} w^{(l)}[i] - \frac{1}{\left|S^{(k)}[i]\right|} \sum_{l \in S^{(k)}[i]} w^{(l)}[i] \tag{221}$$

$$= \left(m + \frac{1}{\left|S^{(j)}[i]\right|} \sum_{l \in S^{(j)}[i] - (S^{(j)}[i] \cap S^{(k)}[i])} (w^{(l)}[i] - m)\right) \tag{222}$$

$$- \left(m + \frac{1}{\left|S^{(k)}[i]\right|} \sum_{l \in S^{(k)}[i] - (S^{(j)}[i] \cap S^{(k)}[i])} (w^{(l)}[i] - m)\right) \tag{223}$$

$$= \frac{1}{\left|S^{(j)}[i]\right|} \sum_{l \in S^{(j)}[i] - S^{(k)}[i]} (w^{(l)}[i] - m) - \frac{1}{\left|S^{(k)}[i]\right|} \sum_{l \in S^{(k)}[i] - S^{(j)}[i]} (w^{(l)}[i] - m) \tag{224}$$

$$\leq \frac{1}{\left|S^{(j)}[i]\right|} \sum_{l \in S^{(j)}[i] - S^{(k)}[i]} (x^{(max)}[i] - m) - \frac{1}{\left|S^{(k)}[i]\right|} \sum_{l \in S^{(k)}[i] - S^{(j)}[i]} (x^{(min)}[i] - m) \tag{225}$$

$$= \frac{\left|S^{(j)}[i] - S^{(k)}[i]\right|}{\left|S^{(j)}[i]\right|} (x^{(max)}[i] - m) + \frac{\left|S^{(k)}[i] - S^{(j)}[i]\right|}{\left|S^{(k)}[i]\right|} (m - x^{(min)}[i]), \tag{226}$$

where the inequality uses Lemma 21. Note that Lemma 21 also implies that $x^{(min)}[i] \leq m \leq x^{(max)}[i]$ since $m$ is the average of some real numbers, all of which are within the range of the values proposed by the honest nodes. Now notice that using Lemma 20 we have

$$\frac{\left|S^{(j)}[i] - S^{(k)}[i]\right|}{\left|S^{(j)}[i]\right|} = \frac{\left|S^{(j)}[i] - S^{(k)}[i]\right|}{\left|S^{(j)}[i] - S^{(k)}[i]\right| + \left|S^{(j)}[i] \cap S^{(k)}[i]\right|} \tag{227}$$

$$\leq \frac{f}{f + \left|S^{(j)}[i] \cap S^{(k)}[i]\right|} \tag{228}$$

$$\leq \frac{f}{f + \varepsilon f} = 1 - \tilde{\varepsilon}, \tag{229}$$

where we used the fact that $\left|S^{(j)}[i] \cap S^{(k)}[i]\right| \geq q - 2f = n - 3f \geq \varepsilon f$, and similarly,

$$\frac{\left|S^{(k)}[i] - S^{(j)}[i]\right|}{\left|S^{(k)}[i]\right|} \leq 1 - \tilde{\varepsilon}. \tag{230}$$

Combining these inequalities with Equation (226), we then obtain

$$\left|y^{(j)}[i] - y^{(k)}[i]\right| \leq (1 - \tilde{\varepsilon}) \left(x^{(max)}[i] - x^{(min)}[i]\right) = (1 - \tilde{\varepsilon}) \Delta^{cw}(\vec{x})[i]. \tag{231}$$

Therefore, we have

$$\Delta^{cw}(\vec{y})[i] \leq (1 - \tilde{\varepsilon}) \Delta^{cw}(\vec{x})[i], \tag{232}$$

which is what we wanted. $\square$

**Remark 5.** *Note that, in the regime $f \ll n$, we can set $\varepsilon \to \infty$, in which case we have $\tilde{\varepsilon} \to 1$. Thus, for any fixed value of $N$, when $f$ is sufficiently small compared to $n$, RB-TM actually achieves asymptotic agreement in only one communication round.*

We have the following corollary regarding RB-TM.

**Corollary 1.** *Under Assumption 6, for any input $N \in \mathbb{N}$, the algorithm RB-TM guarantees Byzantine asymptotic agreement, i.e.,*

$$\Delta_2 \left(\overrightarrow{\text{RB-TM}}_N \circ \overrightarrow{\text{BYZ}}_N(\vec{x})\right) \leq \frac{\Delta_2(\vec{x})}{2^N}. \tag{233}$$

*Proof.* First note that since RB-TM iterates TM, there is actually a sequence of attacks $\overrightarrow{\text{BYZ}}_t$ at each iteration $t \in [T_{\text{RB-TM}}(N)]$. In fact, we have a sequence of families $\vec{y}_t$ defined by $\vec{y}_0 \triangleq \vec{x}$ and $\vec{y}_{t+1} \triangleq \overrightarrow{\text{TM}} \circ \overrightarrow{\text{BYZ}}_t(\vec{y}_t)$ for $t \in [T_{\text{RB-TM}}(N) - 1]$. We eventually have $\overrightarrow{\text{RB-TM}} \circ \overrightarrow{\text{BYZ}}(\vec{x}) = \vec{y}_{T_{\text{RB-TM}}(N)}$.

Note that the previous lemma implies that

$$\Delta^{cw}\left(\vec{y}_{t+1}\right) \leq (1 - \tilde{\varepsilon})\,\Delta^{cw}\left(\vec{y}_t\right). \tag{234}$$

Taking the $\ell_2$ norm on both sides then implies that

$$\Delta_2^{cw}\left(\vec{y}_{t+1}\right) = \left\|\Delta^{cw}\left(\vec{y}_{t+1}\right)\right\|_2 \leq (1 - \tilde{\varepsilon})\left\|\Delta^{cw}\left(\vec{y}_t\right)\right\|_2 = (1 - \tilde{\varepsilon})\,\Delta_2^{cw}\left(\vec{y}_t\right). \tag{235}$$

It follows straightforwardly that

$$\Delta_2^{cw}\left(\vec{y}_{T_{\text{RB-TM}}(N)}\right) \leq (1 - \tilde{\varepsilon})^{T_{\text{RB-TM}}(N)}\,\Delta_2^{cw}\left(\vec{x}\right) \tag{236}$$

$$\leq (1 - \tilde{\varepsilon})^{\frac{(N+1)\ln 2 + \ln\sqrt{h}}{\tilde{\varepsilon}}}\,\Delta_2^{cw}\left(\vec{x}\right) \tag{237}$$

$$= \exp\left(\frac{\ln(1 - \tilde{\varepsilon})}{\tilde{\varepsilon}}\left((N+1)\ln 2 + \ln\sqrt{h}\right)\right)\Delta_2^{cw}\left(\vec{x}\right) \tag{238}$$

$$\leq \exp\left(-(N+1)\ln 2\right)\exp\left(-\ln\sqrt{h}\right)\Delta_2^{cw}\left(\vec{x}\right) \tag{239}$$

$$= \frac{1}{2^{1+N}\sqrt{h}}\Delta_2^{cw}\left(\vec{x}\right), \tag{240}$$

where, in Equation (239), we used $\ln(1 + u) \leq u$ for $u \in (-1, 0]$. We now conclude by invoking Lemma 18, which implies

$$\Delta_2(\vec{y}_{T_{\text{RB-TM}}(N)}) \leq \Delta_2^{cw}(\vec{y}_{T_{\text{RB-TM}}(N)}) \leq 2^{-N}\frac{\Delta_2^{cw}(\vec{x})}{2\sqrt{h}} \leq 2^{-N}\Delta_2\left(\vec{x}\right), \tag{241}$$

which proves that RB-TM achieves asymptotic agreement. $\qquad\square$

We now prove our theorem.

*Proof of theorem 6.* Consider a family $\vec{x}_0 \in \mathbb{R}^{d \cdot h}$. We first focus on coordinate $i \in [d]$ only. We sort the family using a permutation $\sigma$ of $[h]$, so that

$$x_0^{(\sigma(1))}[i] \leq x_0^{(\sigma(2))}[i] \leq \ldots \leq x_0^{(\sigma(h-1))}[i] \leq x_0^{(\sigma(h))}[i]. \tag{242}$$

Now denote $q_j = \left|Q^{(j)}\right|$, and $\vec{z} = \overrightarrow{\text{BYZ}}_0^{(j)}(\vec{x}_0) = \vec{w}_0^{(Q^{(j)})} \in \mathbb{R}^{d \cdot q_j}$ the result of a Byzantine attack. Again, we sort the vectors of this family, using a permutation $\tau$ of $[q_j]$, so that

$$z^{(\tau(1))}[i] \leq z^{(\tau(2))}[i] \leq \ldots \leq z^{(\tau(q_j-1))}[i] \leq z^{(\tau(q_j))}[i]. \tag{243}$$

Now, denoting $y \triangleq \text{TM}(\vec{z})$, we note that

$$y[i] = \frac{1}{q_j - 2f}\sum_{k=1}^{q_j - 2f} z^{(\tau(f+k))}[i]. \tag{244}$$

Moreover, note that there are $f + k - 1$ values of $\vec{z}$ that are smaller than $z^{(\tau(f+k))}[i]$. These can include $f$ Byzantine vectors. But the remaining $k - 1$ values must then come from the family of honest vectors. Yet, the $k - 1$ smallest vectors of this family are $x_0^{(\sigma(1))}[i], \ldots, x_0^{(\sigma(k-1))}[i]$. But then, $z^{(\tau(f+k))}[i]$ will have to take a value on the right of $x_0^{\sigma(k-1)}[i]$ in the list of honest vectors, which corresponds to saying that

$$\forall k \in [q_j - f],\ z^{(\tau(f+k))}[i] \geq x_0^{(\sigma(k))}[i]. \tag{245}$$

But then, we know that

$$y[i] \geq \frac{1}{q_j - 2f}\sum_{k=1}^{q_j - 2f} x_0^{(\sigma(k))}. \tag{246}$$

As an immediate corollary, we see that $y[i] \geq x_0^{(\sigma(1))}[i]$, which also implies that

$$x_0^{(\sigma(k))}[i] \leq x_0^{(\sigma(1))}[i] + \max_{l \in [h]}\left(x_0^{(\sigma(l))}[i] - x_0^{(\sigma(1))}[i]\right) \leq y[i] + \Delta^{cw}(\vec{x}_0)[i]. \tag{247}$$

But now notice that

$$\bar{x}_0[i] = \frac{1}{h} \sum_{k=1}^{h} x_0^{(\sigma(k))} = \frac{1}{h} \sum_{k=1}^{q_j-2f} x_0^{(\sigma(k))} + \frac{1}{h} \sum_{k=q_j-2f+1}^{h} x_0^{(\sigma(k))} \tag{248}$$

$$\leq \frac{1}{h} \left( (q_j - 2f)y[i] \right) + \frac{1}{h} \sum_{k=q_j-2f+1}^{h} \left( y[i] + \Delta^{cw}(\vec{x}_0)[i] \right) \tag{249}$$

$$= y[i] + \frac{h - q_j + 2f}{h} \Delta^{cw}(\vec{x}_0)[i]. \tag{250}$$

Similarly, we can also prove that $\bar{x}_0[i] \geq y[i] - \frac{h-q_j+2f}{h}\Delta^{cw}(\vec{x}_0)[i]$, which implies that

$$|y[i] - \bar{x}_0[i]| \leq \frac{h - q_j + 2f}{h} \Delta^{cw}(\vec{x}_0)[i] \leq \frac{2f}{h} \Delta^{cw}(\vec{x}_0)[i], \tag{251}$$

where we used the fact that $q_j \geq q = h$. Thus, $\|y - \bar{x}_0\|_2 \leq \frac{2f}{h} \|\Delta^{cw}(\vec{x}_0)\|_2 = \frac{2f}{h}\Delta_2^{cw}(\vec{x}_0)$. In fact, more generally, we showed that, for any Byzantine attack $\overrightarrow{\mathrm{BYZ}}_0^{(j)}$, we have

$$\mathrm{TM} \circ \overrightarrow{\mathrm{BYZ}}_0^{(j)}(\vec{x}_0) \in Y_0 = \bar{x}_0 + \frac{2f}{h} \prod_{i \in [d]} \left[ -\Delta^{cw}(\vec{x}_0)[i], +\Delta^{cw}(\vec{x}_0)[i] \right]. \tag{252}$$

Yet Lemma 21 shows that any such parallelepiped was stable under application of TM despite Byzantine attacks. Thus, for any iteration $t \geq 1$, we still have $x_t^{(j)} \in Y_0$, which then guarantees that

$$\|\bar{x}_t - \bar{x}_0\|_2 = \left\| \frac{1}{h} \sum_{j \in [h]} \left( x_t^{(j)} - \bar{x}_0 \right) \right\|_2 \leq \frac{1}{h} \sum_{j \in [h]} \left\| x_t^{(j)} - \bar{x}_0 \right\|_2 \tag{253}$$

$$\leq \frac{1}{h} \sum_{j \in [h]} \frac{2f}{h} \|\Delta^{cw}(\vec{x}_0)\|_2 = \frac{2f}{h}\Delta_2^{cw}(\vec{x}_0). \tag{254}$$

Lemma 18 then guarantees $\Delta_2^{cw}(\vec{x}_0) \leq 2\sqrt{h}\Delta_2(\vec{x}_0)$. By noting that RB-TM corresponds to iterating TM, we conclude that RB-TM achieves $\frac{4f}{\sqrt{h}}$-averaging agreement.

Now we show that for $n \leq 3f$, no algorithm can achieve Byzantine averaging agreement. If $n \leq 3f$, then $h = n - f \leq 2f$. Thus honest nodes can be partitioned into two subsets of cardinals at most $f$. In particular, for any subset, Byzantine nodes can block all messages coming from the other subset. Any subset would thus only hear from nodes of the subset and from the Byzantine nodes.

Assume now by contradiction that AVG achieves Byzantine-resilience averaging agreement for $n \leq 3f$. Note that we then have $q \leq 2f$. As a result $q - f \leq f$. But as a result, if Byzantines send $\vec{z} = z \star f$ to all honest nodes, quasi-unanimity (Lemma 17) applies, which means that all honest nodes must output $z$.

But this hold for any value $z$ chosen by the Byzantine nodes. Clearly, this prevents averaging. Thus AVG fails to achieve averaging agreement. $\square$