# OpenReview forum: "Collaborative Learning in the Jungle (Decentralized, Byzantine, Heterogeneous, Asynchronous and Nonconvex Learning)"
_NeurIPS.cc/2021/Conference — NeurIPS 2021 Poster_

### Official Review · Reviewer_A4Yt · 2021-07-14

**Rating:** 7
**Confidence:** 3

**Summary:**

The authors present a study on Byzantine collaborative learning. In it, n nodes try to learn from each other’s data collectively and some f < n nodes can be adversarial to the learning process and are defined Byzantine. Such problem of collaborative learning is in a fully decentralised, Byzantine, heterogeneous and asynchronous environment with non-convex loss functions.
In the first part, the authors formulate the collaborative learning problem; They then present an equivalence between collaborative learning and a problem called averaging agreement via two reductions from the former to the latter and vice versa.
Their reduction proves useful to draw impossibility results and optimal algorithms for collaborative learning through the averaging agreement problem.
Therefore, they then move on to define a Byzantine collaborative learning algorithm called LEARN that allows to solve a Byzantine collaborative learning problem given an averaging agreement problem.
In the last section, the authors present and discuss some empirical evaluations.




**Limitations And Societal Impact:**

Seems adequate.

**Main Review:**

General Comments
The paper is very precise accurate, and its structure is very clear. The double-ended reduction is quite interesting and suggests it could become a powerful tool.
The notation is a bit cumbersome and reminding what some values represent somewhere along the paper could help improve the readability of the work.
A couple of key concepts are mentioned and used but not well-presented nor described. For example, the averaging agreement protocol is barely defined and it could be better appreciated with more explanations.

General Feeling
The paper is solid and shows attention to the details, the modifications suggested are intended to correct minor lacks. I recommend acceptance.

Detailed Notes
-	Theorem 5 the second sentence suggests to be a corollary. Consider making it one.
-	Figure 1 is very small and in this way hard to appreciate, I suggest making it bigger. Possibly not framed in text if not requested by the format guidelines
-	Despite putting the reference to the definition of Byzantine problems, a brief sentence could be helpful to recall what the authors refer to with the term Byzantine.
-	I cannot seem to find a real description of what averaging agreement is, nor a citation to works having it
-	Page 3 line 106 “in order of billions…”  “in the order of billions…”

**Time Spent Reviewing:**

6h

---

> ### Author Response · Authors · 2021-08-10
> **Official Response to Reviewer A4Yt about Paper3354**
>
> We thank the reviewer for their useful comments. Below, we discuss the reviewer’s concerns.
>
> > For example, the averaging agreement protocol is barely defined and it could be better appreciated with more explanations.
>
> We agree. We added the following remarks after introducing the averaging agreement problem:
> “In Section 4, we will present two solutions to the averaging agreement problem. These solutions typically involve several rounds. At each round, each node sends its current vector to all other nodes. Then, once a node has received sufficiently many vectors, it will execute a robust mean estimator to these vectors, the output of which will be their starting vector for the next round. The nodes then halt after a number of rounds dependent on the parameter N.”
>
> > Theorem 5 the second sentence suggests to be a corollary. Consider making it one.
>
> To clarify, Theorem 5 proves a lower bound on the value of C that any averaging agreement algorithm can guarantee. We modified it to clearly highlight this.
>
> We moved the implications on the near-optimality of MDA to the main text, which is indeed a corollary.
>
> > Figure 1 is very small and in this way hard to appreciate, I suggest making it bigger. Possibly not framed in text if not requested by the format guidelines
>
> Done.
>
> > Despite putting the reference to the definition of Byzantine problems, a brief sentence could be helpful to recall what the authors refer to with the term Byzantine.
>
> We added the phrase “i.e., they can behave arbitrarily maliciously, to confuse the system”.
>
> > I cannot seem to find a real description of what averaging agreement is, nor a citation to works having it
>
> To the best of our knowledge, our work is the first to define the averaging agreement problem. We hope that our paragraph (mentioned above) on the solutions we propose gives the reader insights into its challenges.
>
> > Page 3 line 106 “in order of billions…”  “in the order of billions…”
>
> Done. Thank you for catching this typo.

---

> > ### Comment · Reviewer_A4Yt · 2021-09-10
> > **Ack**
> >
> > I acknowledge that I have read the rebuttal.

---

### Official Review · Reviewer_RG8y · 2021-07-14

**Rating:** 6
**Confidence:** 3

**Summary:**

The paper presents algorithms for collaborative learning which are distributed and robust wrt Byzantine nodes as long as these do not exceed either 1/3 or 1/6 of the total number of nodes.  This result is achieved by proving the equivalence of two concepts the authors introduce, C-learning and C-averaging where C is a constant.  Computational results are provided to support the claims of the paper.

**Limitations And Societal Impact:**


It is hard to see where the authors identify key limitations of this paper often buried under 1/2 dozen assumptions made.  This is quite serious as there appears to be a large amount of averaging going on in the cross-node learning process necessitated to filter out the influence of the Byzantine nodes which is not explicitly localized and stated except implicitly within the 1/2 dozen assumptions.  The authors can start by explaining briefly where and why each assumption is needed.

**Main Review:**


The paper tackles an important problem in distributed learning, namely, with the existence of Byzantine nodes, how does one construct cross-node learning algorithms which automatically and effectively filter out the influence of the unknown miscreant nodes and leverage neighbor node learning?    The writing of this paper is generally clear but occasionally there are lapses of clarity which turn out to be quite essential for following the basic narrative of this paper.  For example, the key concepts of C-leaning and C-averaging are defined only with reference to honest nodes but each node has to update its gradient vector and \theta estimates without such explicit knowledge.  So what is the exact procedure each nodes follows?  Is that what is shown as Algorithm 1 on line 240?  What's BYZ in steps 6 and 8 in this algorithm?   What does the equivalence of C-learning and C-averaging -- which involve honest nodes only -- has to say about the impact of Byzantine nodes on estimate of the gradient communicated to other nodes?   Remark 2 seems to mean T_{LEARN} scales like 1/\delta^3 and since batch size (see line 251) is set to t for t<=T_1 and equal to T_1 thereafter, nodes potentially have to use enormous batch sizes.  Are super large batch sizes needed to dilute the contribution of Byzantine nodes?  Is it then implicitly assumed that each node effectively has unlimited data from its own distribution?  Where do the coefficients 1/3 ( n > 3f) and 1/6 (n>6f) come from?  How does one ensure Assumption 5, which gives rise to 6 in n> 5f, holds in any given setting?  What happens if the data for all the honest nodes come from the same distribution, in other works when there is no heterogeneity?  In this case K=0 but it seems this paper has nothing to offer in this case, per statement on line 216.

**Time Spent Reviewing:**

4

---

> ### Author Response · Authors · 2021-08-10
> **Official Response to Reviewer RG8y about Paper3354**
>
> We thank the reviewer for their useful remarks. Below, we discuss the reviewer’s concerns.
>
> > For example, the key concepts of C-leaning and C-averaging are defined only with reference to honest nodes but each node has to update its gradient vector and \theta estimates without such explicit knowledge. So what is the exact procedure each nodes follows? Is that what is shown as Algorithm 1 on line 240? What's BYZ in steps 6 and 8 in this algorithm?
>
> Similar to other works on Byzantine resilience, the goal of our paper is to guarantee that the honest nodes will converge to a desirable state despite the arbitrary behavior of the Byzantine nodes and without knowing which nodes are Byzantine and which nodes are honest.
>
> Now, because we also prove lower bounds, we considered very general definitions of C-learning and C-averaging that allow for arbitrary forms of communications. In particular, this is why we cannot precisely define what Byzantine attacks are most harmful and ought to be considered: these attacks depend on the algorithm LEARN and AVG that are actually considered. But no matter what Byzantine nodes do and how they attack, our guarantees still hold.
>
> In the case of Algorithm 1, the only communication phases are those that leverage the C-averaging oracle. Algorithm 1 thus contains the exact procedure an honest node must execute, up to the definition of the C-averaging oracle. It can be instantiated for any particular C-averaging algorithm.
>
> To be more precise, we rewrote these two lines as:
> * $\gamma_t \leftarrow AVG_{N(t)} (\vec g_t, BYZ)$,
> * $\theta_{t+1} \leftarrow AVG_1 (\vec \theta_{t+1/2}, BYZ)$,
>
> and added the following explanation:
> “We stress that on steps 6 and 8, when the averaging agreement algorithm AVG is called, the Byzantines can adopt any procedure BYZ, which consists in sending any message to any node at any point based on any information in the system, and in delaying for any amount of time any message sent by any honest node. Note that, apart from this, all other steps of Algorithm 1 are purely local operations.”
>
> In Section 4, two algorithms AVG are described:
> * MDA leverages the classical well-known asynchronous round-based solution of Bracha (1987), which is described in the second paragraph of Section 2.1, where at each round t, each node sends their time-t vector, and waits until they receive q time-t vectors from different nodes. MDA is then executed on the collected vectors.
> * RB-TM leverages the well-known reliable broadcast algorithm and a witness mechanism, to prevent Byzantines from successfully delivering different messages to different nodes. Again, a round-based system is executed, to reduce the diameter of honest nodes’ vectors at each round.
>
> > What does the equivalence of C-learning and C-averaging -- which involve honest nodes only -- has to say about the impact of Byzantine nodes on estimate of the gradient communicated to other nodes?
>
> The difficulty of C-learning and C-averaging is precisely to guarantee that Byzantine nodes cannot be harmful to the honest nodes, despite the fact that Byzantine nodes can behave arbitrarily.
> Our equivalence proves that any solution to one problem can be used to construct a solution to the other problem.
>
> > Remark 2 seems to mean T_{LEARN} scales like 1/\delta^3 and since batch size (see line 251) is set to t for t<=T_1 and equal to T_1 thereafter, nodes potentially have to use enormous batch sizes. Are super large batch sizes needed to dilute the contribution of Byzantine nodes?
>
> In short, yes.
>
> As empirically evidenced by [4], a large variance of the gradient sampling makes collaborative learning more vulnerable to Byzantine attacks. Yet, this variance is inversely proportional to the batch size. Thus, Byzantine resilience seems to require a sufficiently large batch size.
>
> Our paper provides a formalization of this fundamental tradeoff between the batch size and Byzantine resilience. Indeed, Theorem 5 shows that no algorithm can guarantee C-averaging (for C too small), which means that its error in estimating the mean is necessarily proportional to the diameter of honest vectors.
>
> Yet, especially in high dimension d, this diameter is very large for small batch size b. In fact, it will typically be of the order $\sqrt{d/b}$ (say, for normally distributed gradients). Thus, the batch size must eventually be large enough.
>
> Remark 1 highlights the fact that making this batch size grow linearly with the number of iterations is a natural way to proceed. Evidently, this batch size could also be set large enough from the first iteration, but we provide an intuitive discussion on line 175 on why letting the batch size grow seems preferable.
>
> > Is it then implicitly assumed that each node effectively has unlimited data from its own distribution?
>
> No. As explained in the opening of Section 2.2, our results hold equally well for statistical risk (with unlimited data from a local distribution) and for empirical risk (where a zero-variance is achieved by computing the exact gradient).
>
> Besides, our algorithm LEARN terminates in a finite number of rounds; thus, even for the case of statistical risk, we only need to sample a bounded number of data points (which depends on the parameter $\delta$).
>
> > Where do the coefficients 1/3 ( n > 3f) and 1/6 (n>6f) come from?
>
> They are precisely derived in the proofs, given in the Supplementary Material. In particular, $1/3$ matches the classical result of Abraham, Amit and Dolev (2004) in dimension 1. Intuitively, this is because, using reliable broadcast and a witness mechanism, we can guarantee that any two nodes will have at least q message in common, which can be used to achieve approximate agreement.
>
> The $1/6$ is trickier to explain directly, but it essentially results from the fact that when f is sufficiently small, two nodes will receive sufficiently many vectors in common, so that the Byzantine nodes cannot attack MDA sufficiently to prevent approximate agreement. In the supplementary material (Proposition 1 on line 816), we also prove that $n>6f$ is actually necessary for correctness of MDA.
>
> > How does one ensure Assumption 5, which gives rise to 6 in n> 5f, holds in any given setting?
>
> First, note that such assumptions are common to guarantee Byzantine resilience in distributed computing. Nevertheless, the reviewer’s question is very important in practice. A thorough auditing of nodes is arguably needed to assess the individual risks of a node being malicious or hacked. In highly controlled environments, maybe a datacenter, or with machines regularly controlled, $n > 6f$ may be reasonable. In less controlled environments, such as using users’ phones, more resilient (and usually computationally costly) algorithms are needed, e.g. RB-TM.
>
> > What happens if the data for all the honest nodes come from the same distribution, in other works when there is no heterogeneity? In this case K=0 but it seems this paper has nothing to offer in this case, per statement on line 216.
>
> Our paper addressed the homogeneous case in Section 3.3. There, we proved that our reduction successfully outputs a solution, whose gradient is arbitrarily small. We even propose an optimized algorithm for this specific case in the Supplementary Material (Algorithm 2).
>
> > It is hard to see where the authors identify key limitations of this paper often buried under 1/2 dozen assumptions made. This is quite serious as there appears to be a large amount of averaging going on in the cross-node learning process necessitated to filter out the influence of the Byzantine nodes which is not explicitly localized and stated except implicitly within the 1/2 dozen assumptions. The authors can start by explaining briefly where and why each assumption is needed.
>
> Our distributed computing assumptions are extremely general and classical. They are detailed in Section 2.1. More specific assumptions about f and n are precisely given just before the statement of the theorems that require them (e.g., Theorems 4 and 6).
>
> On the machine learning front, we clearly stated our assumptions in Section 2.2. Again, the first three assumptions are very classical assumptions in machine learning. Assumption 4 is the only non standard assumption. We provided a discussion immediately after its statement, detailing why it is reasonable.
>
> Indeed, our reduction LEARN requires many averaging steps. But actually, our key contribution is to abstract away all these averaging steps, by considering an oracle AVG that contains all the communication steps, and by only leveraging the averaging agreement guarantees of this oracle. This allowed us to cleanly prove the correctness of LEARN.
>
> The correctness of specific algorithms AVG was then formally proved in Section 4.

---

### Official Review · Reviewer_8fQX · 2021-07-16

**Rating:** 5
**Confidence:** 4

**Summary:**

This paper considers the Byzantine-robust learning problem in a decentralized network. The authors discuss the connection between Byzantine-robust learning and averaging agreement.

**Limitations And Societal Impact:**

No comment.

**Main Review:**

The authors claim that this paper addresses the problem of Byzantine-robust learning for the decentralized, asynchronous and heterogeneous setting. However, several issues must be clarified to avoid misunderstanding.
- The decentralized topology seems to be a complete graph. This is different to, and relatively simpler than the incomplete graph that is of interest to the machine learning community.
- The asynchronous communication model assumes the malicious nodes can delay transmissions to the honest nodes. Intuitively the malicious nodes would like to send messages as fast as possible so as to maximize the effects of attacks. Therefore, this asynchronous setting can reduce to the worst case: among the q messages received by any honest node, f of them are malicious.
- Heterogeneity is a big challenge to Byzantine-robustness. This paper discusses its impact, but does not show how to address this issue.

I understand that the authors focus on the theoretical part of Byzantine-robustness, but sufficient numerical experiments are necessary for a NeurIPS paper. At least, the authors should compare the proposed methods with the existing decentralized Byzantine-robust learning algorithms. The authors should also show more performance metrics, in addition to slowdown.

The proposed MDA method is not new. Please cite proper references and give some discussions.


**Time Spent Reviewing:**

6

---

> ### Author Response · Authors · 2021-08-10
> **Official Response to Reviewer 8fQX about Paper3354**
>
> We thank the reviewer for their useful remarks. Below, we discuss the reviewer’s concerns.
>
> >The decentralized topology seems to be a complete graph. This is different to, and relatively simpler than the incomplete graph that is of interest to the machine learning community.
>
> Indeed, we consider that any node must communicate with every other node, potentially by letting intermediary nodes relay messages. In practice, as the number of nodes grows, this can be very costly. Future work should investigate topology-optimized distributed learning schemes. We believe that studying complete graphs is however a first important step.
>
> Reducing communication capabilities (i.e. incomplete graphs) will however make it harder to make the learning Byzantine resilient. In particular, if any honest node is surrounded by a majority of Byzantine nodes, then it is hopeless to make it learn the same vector as all other honest nodes, and thus to achieve collaborative learning.
>
> Additionally, our work provides a very good intuition about what can/cannot be achieved considering a more general topology. For instance, clearly, our impossibility results hold for a more general incomplete graph.
>
> > The asynchronous communication model assumes the malicious nodes can delay transmissions to the honest nodes. Intuitively the malicious nodes would like to send messages as fast as possible so as to maximize the effects of attacks. Therefore, this asynchronous setting can reduce to the worst case: among the q messages received by any honest node, f of them are malicious.
>
> Indeed. Note that, in addition, the worst case is achieved when Byzantine nodes maliciously select the q-f honest messages that will be received by the honest node. This observation is widely used in our lower bound proofs.
>
> > Heterogeneity is a big challenge to Byzantine-robustness. This paper discusses its impact, but does not show how to address this issue.
>
> Handling heterogeneity is actually a key contribution of our paper.  On the one hand, our reduction successfully outputs a solution whose optimality (measured by the norm of the gradient of the honest loss) is provably bounded by a constant times a squared measure of heterogeneity (parameter $K$ in Definition 1).
>
> On the other hand, and perhaps most interestingly, we prove that no algorithm can provide a better guarantee than $C^2$ times the square heterogeneity (as we defined it), for $C < (h+2f-q)/h$. This results from our converse reduction and Theorem 5.
>
> Put differently, our paper proves that, in a precise sense, heterogeneity is optimally addressed by our reduction.
>
> > I understand that the authors focus on the theoretical part of Byzantine-robustness, but sufficient numerical experiments are necessary for a NeurIPS paper. At least, the authors should compare the proposed methods with the existing decentralized Byzantine-robust learning algorithms. The authors should also show more performance metrics, in addition to slowdown.
>
> Our contribution is mainly theoretical, and we intentionally used most of the paper to present the theoretical results as clearly as possible. Nevertheless, we agree that more numerical assessments would be desirable. Although we believe that the theoretical contribution is a fundamental first step,  we will run additional experiments by tracking the loss and the diameter of honest nodes during training, under attack from one node.
>
> > The proposed MDA method is not new. Please cite proper references and give some discussions.
>
> What we call the minimum diameter averaging (MDA) approach was introduced in [33] by Rousseeuw in the context of robust statistics and optimal breakdown point. It was referred to as the minimal volume ellipsoid, then by [14] in the context of robust machine learning. We added this in the paper where MDA is first mentioned.

---

> > ### Comment · Reviewer_8fQX · 2021-09-10
> > **ack**
> >
> > I acknowledge that I have read the rebuttal.

---

### Official Review · Reviewer_cDim · 2021-07-22

**Rating:** 7
**Confidence:** 3

**Summary:**

The paper considers collaborative learning in an (asynchronous) decentralized setup in the presence of Byzantine nodes. The authors define two problems: Byzantine-robust collaborative learning and Byzantine-robust ‘averaging agreement’. The main contribution is to show an equivalence between these two problems. Then, the paper proposes two algorithms for averaging agreement, which yield Byzantine-robust collaborative learning solutions. Finally, the paper shows that when the number of Byzantine nodes is larger than 1/3, then no algorithm can achieve Byzantine averaging agreement.

**Limitations And Societal Impact:**

The authors have discussed potential negative societal impact. It will be helpful to discuss limitations of the proposed averaging agreement algorithms: e.g., runtime of MDA is exponential in q when q is comparable to f, and trimmed-mean requires reliable broadcast assumption.

**Main Review:**

*Originality and Quality:* The notion of Byzantine averaging agreement and the equivalence result are interesting. As mentioned in the paper, the equivalence result is powerful, as it allows one to obtain impossibility results for collaborative learning. On the other hand, I have three major concerns.

1. The collaborative learning definition (Definition 1) warrants more justification. Works on Byzantine-robust distributed learning typically present learning algorithms which converge to a critical point of the (non-convex) statistical loss function. Is \bar{theta} serving the same purpose according to (4)? It will be important to add a discussion here. Without comparing to standard collaborative/distributed learning setups and objectives, it is difficult to appreciate the definition.

2. The paper assumes that there is a predefined common seed. How is the common seed generated in the presence of Byzantine nodes? It is helpful to give more details on this.

3. As mentioned in Remark 1, batch size needs to grow linearly with the number of iterations for the proof of Theorem 1 to hold. This seems to be quite restrictive from a practical perspective. Is this an artifact of the proof techniques or is it more fundamental? It would be important to add more details on this.

Apart from these, another question is about the reliable broadcast model. Is the reliable broadcast decentralized model equivalent to the (trusted) parameter-server-based model (operationally, both the models seem to be the same)? It will be helpful to compare and contrast the two models.

*Clarity:* The paper is generally well-written, but the quality can be improved. Please find below some comments.

1. It will be helpful to comment on how the formulation (in terms of the objective function) of collaborative learning compares with [36] (other than convex loss and synchrony assumption). Also, how does the model compare with that in [13]?

2. It will be important to explicitly define the notation in lines 6 and 8 of Algorithm 1. How is BYZ(.) defined?

3. According to Definition 2, AVG outputs a family of vectors. As per (10), MDA outputs a single vector. I may be missing something here, but it would clearly be helpful to provide more details.

4. It will be helpful to formally define \bar{theta}. Assumption 5 should be stated in Theorem 4.

*Significance:* Unlike Byzantine-robust distributed learning (with a parameter server), which has received significant research attention recently, decentralized learning with Byzantine-resilience has received much less research attention. The asynchronous and omniscient adversary setup considered seems a bit too strong for practice, but it is intellectually interesting. The notion of averaging agreement and the equivalence result are quite interesting. I am willing to reconsider my score if the authors can address the major concerns and improve the clarity of exposition.

**Time Spent Reviewing:**

5

---

> ### Author Response · Authors · 2021-08-10
> **Official Response to Reviewer cDim about Paper3354**
>
> > The asynchronous and omniscient adversary setup considered seems a bit too strong for practice, but it is intellectually interesting. The notion of averaging agreement and the equivalence result are quite interesting. I am willing to reconsider my score if the authors can address the major concerns and improve the clarity of exposition.
>
> We thank the reviewer for their perspicacious review and for their openness. Below, we discuss the reviewer’s concerns.
>
> > The collaborative learning definition (Definition 1) warrants more justification. Works on Byzantine-robust distributed learning typically present learning algorithms which converge to a critical point of the (non-convex) statistical loss function. Is \bar{theta} serving the same purpose according to (4)? It will be important to add a discussion here. Without comparing to standard collaborative/distributed learning setups and objectives, it is difficult to appreciate the definition.
>
> Our goal is indeed to have $\bar{\theta}$ converge to a critical point of the average of honest loss functions, and we prove that our algorithm achieves this in the homogeneous case. However, our impossibility results prove that this cannot be guaranteed in the presence of Byzantine nodes in the heterogeneous case.
>
> Indeed, the contrapositive of Theorem 2, combined with the impossibility of averaging agreement for a very small $C$ (Theorem 5), proves the impossibility of collaborative learning for $C$ too small. Put differently, no algorithm can output a solution for which the norm of the gradient is guaranteed to be less than $C^2 K^2$.
>
> > The paper assumes that there is a predefined common seed. How is the common seed generated in the presence of Byzantine nodes? It is helpful to give more details on this.
>
> The seed can be hard-coded as part of the starting algorithm (like the rest of the instructions of the algorithm). Note that only the honest nodes are assumed to use it. We added this remark to the paper.
>
> Note that, alternatively, assuming partial synchrony in the initialization phase, the seed could have been shared using any consensus algorithm.
>
> > As mentioned in Remark 1, batch size needs to grow linearly with the number of iterations for the proof of Theorem 1 to hold. This seems to be quite restrictive from a practical perspective. Is this an artifact of the proof techniques or is it more fundamental? It would be important to add more details on this.
>
> The short answer is: we believe that increasing the batch size is fundamentally required to achieve convergence in the presence of Byzantine nodes.
>
> Indeed, as empirically discussed in [4], a large variance of the gradient sampling makes collaborative learning more vulnerable to Byzantine attacks. Yet, this variance is inversely proportional to the batch size. Thus, Byzantine resilience seems to require a sufficiently large batch size.
>
> Our paper provides a formalization of this fundamental tradeoff between the batch size and Byzantine resilience. Indeed, Theorem 5 shows that no algorithm can guarantee C-averaging (for C too small), which means that its error in estimating the mean is necessarily proportional to the diameter of honest vectors.
>
> Yet, especially in high dimension $d$, this diameter is very large for small batch size $b$. In fact, it will typically be of the order $\sqrt{d/b}$ (say, for normally distributed gradients). Thus, the batch size must eventually be large enough.
>
> Remark 1 highlights the fact that making this batch size grow linearly with the number of iterations is a natural way to proceed. Evidently, this batch size could also be set large enough from the first iteration, but we provide an intuitive discussion on line 175 on why letting the batch size grow seems preferable.
>
> > Apart from these, another question is about the reliable broadcast model. Is the reliable broadcast decentralized model equivalent to the (trusted) parameter-server-based model (operationally, both the models seem to be the same)? It will be helpful to compare and contrast the two models. (This one was missing)
>
> Note that reliable broadcast is not a different model. All over the paper, we consider a fully distributed asynchronous model, presented in Section 2.1. Reliable broadcast is a primitive that can be implemented in this model to ensure some convenient properties [1] (there are several algorithms for implementing this primitive, e.g.,  [1]). Implementing this primitive first makes the algorithms implemented on top of it simpler than if they were implemented from scratch.
>
> In general, our model is not equivalent to one with a trusted server, since our model does not consider any trusted node. Nevertheless, our key finding is that, for the purpose of learning, we do not fundamentally lose in terms of Byzantine resilience and convergence guarantee, with respect to an approach with a trusted node, albeit we require more communications.
>
> > It will be helpful to comment on how the formulation (in terms of the objective function) of collaborative learning compares with [36] (other than convex loss and synchrony assumption). Also, how does the model compare with that in [13]?
>
> In [36], the authors assume that the local loss functions have a finite-sum structure, and they use the exact (not stochastic) gradient of the local losses in their algorithm. They also assume the data distributions of nodes are the same (homogeneous). Thus, they can guarantee convergence to the exact solution without being affected by our impossibility result. As mentioned by the reviewer, strong convexity and synchrony are two other notable differences.
>
> Similarly, the main difference with [13] is that we consider a heterogeneous environment. There are also some important technical differences (e.g. Byzantine model, network assumption), which make our paper more general.
>
> > It will be important to explicitly define the notation in lines 6 and 8 of Algorithm 1. How is BYZ(.) defined?
>
> The reviewer is right and we added the following explanations:
> “We stress that on steps 6 and 8, when the averaging agreement algorithm AVG is called, the Byzantine nodes can adopt any procedure BYZ, which consists in sending any message to any node at any point based on any information in the system, and in delaying for any amount of time any message sent by any honest node. Note that, apart from this, all other steps of Algorithm 1 are purely local operations.”
>
> We also rewrote these two lines as:
> * $\gamma_t \leftarrow AVG_{N(t)} (\vec g_t, BYZ)$.
> * $\theta_{t+1} \leftarrow AVG_1 (\vec \theta_{t+1/2}, BYZ)$.
>
> > According to Definition 2, AVG outputs a family of vectors. As per (10), MDA outputs a single vector. I may be missing something here, but it would clearly be helpful to provide more details.
>
> The MDA algorithm (also RB-TM) is executed on all nodes in the system. For each node, the algorithm takes a family of vectors as input and outputs a single vector, but the output vector for node 1 is not necessarily the same as the output vector for node 2. Thus the set of all outputs will be a family of vectors.
>
> > It will be helpful to formally define $\bar{theta}$.
>
> Done. We rewrote the informal definition of collaborative learning as follows:
> “collaborative learning consists in minimizing the average $\bar{\mathcal L} (\bar{\theta}) \triangleq \frac{1}{h} \sum_{j \in [h]} \mathcal L^{(j)} (\bar{\theta})$ of local losses at the average $\bar{\theta} \triangleq \frac{1}{h} \sum_{j \in [h]} \theta^{(j)}$, while guaranteeing that the honest nodes' parameters have a small diameter.”
>
> > Assumption 5 should be stated in Theorem 4.
>
> Done.
>
> > The authors have discussed potential negative societal impact. It will be helpful to discuss limitations of the proposed averaging agreement algorithms: e.g., runtime of MDA is exponential in q when q is comparable to f, and trimmed-mean requires reliable broadcast assumption.
>
> Done. We added: “We also note that the computation time of MDA grows exponentially with $q$, when $f$ is a constant fraction of $q$.”
>
> We stress however that trimmed-mean does not require additional network assumption. Moreover, it can be shown to achieve Byzantine averaging agreement even without the use of reliable broadcast, assuming a smaller fraction of Byzantines.

---

### Decision · Program_Chairs · 2021-09-27

**Decision:**

Accept (Poster)

**Comment:**

This paper proposes a Byzantine-robust fully decentralized learning approach under data heterogeneity.

The reviews were initially quite mixed. After reading the author response, several reviewers updated their score. Some concerns about the role/relevance of using large batch sizes, as well as the lack of insights on what robustness can be achieved in the heterogeneous setting. However, the reviewers appreciated the response and recognize the relevance of the problem, approach and evaluation.

Therefore, the paper is accepted. The authors are strongly encouraged to incorporate suggestions and additional details from the discussion, in particular to improve the clarity of the presentation and give more intuitions about the approach.